# Progress in the Development of Detection Strategies Based on Olfactory and Gustatory Biomimetic Biosensors

**DOI:** 10.3390/bios12100858

**Published:** 2022-10-11

**Authors:** Yating Chen, Liping Du, Yulan Tian, Ping Zhu, Shuge Liu, Dongxin Liang, Yage Liu, Miaomiao Wang, Wei Chen, Chunsheng Wu

**Affiliations:** 1Institute of Medical Engineering, Department of Biophysics, School of Basic Medical Sciences, Health Science Center, Xi’an Jiaotong University, Xi’an 710061, China; 2Key Laboratory of Environment and Genes Related to Diseases, Xi’an Jiaotong University, Ministry of Education of China, Xi’an 710061, China

**Keywords:** detection strategies, olfactory and gustatory, biomimetic biosensors

## Abstract

The biomimetic olfactory and gustatory biosensing devices have broad applications in many fields, such as industry, security, and biomedicine. The development of these biosensors was inspired by the organization of biological olfactory and gustatory systems. In this review, we summarized the most recent advances in the development of detection strategies for chemical sensing based on olfactory and gustatory biomimetic biosensors. First, sensing mechanisms and principles of olfaction and gustation are briefly introduced. Then, different biomimetic sensing detection strategies are outlined based on different sensing devices functionalized with various molecular and cellular components originating from natural olfactory and gustatory systems. Thereafter, various biomimetic olfactory and gustatory biosensors are introduced in detail by classifying and summarizing the detection strategies based on different sensing devices. Finally, the future directions and challenges of biomimetic biosensing development are proposed and discussed.

## 1. Introduction

Olfaction and gustation are two of the most important senses in mammals. Olfaction, as a physiological sense, can perceive chemical odorant molecules in the environment. The olfactory system can discriminate different types of odorant molecules with extremely high sensitivity, specificity, and the wide response range [1,2,3,4]. In mammalian olfactory systems, even different odorant molecules with minor molecular structure differences can cause significant changes in odorants perception and enable the real-time identification of odorants at the trace level, which can be attributed to the olfactory receptor and olfactory coding system [3,4,5,6]. Olfactory receptor genes are one of the largest gene families, accounting for approximately 1% of the chromosomal gene pool [7,8,9,10].

As another important biological system, the gustatory system provides five basic tastes: sour, sweet, bitter, salty, and umami [9], which cover all flavors we can taste and identify in real time with high performance. The gustatory system is also important for biological survival by discriminating the harmful substances. Therefore, it is quite crucial to evaluate odorants and tastants for many applications. For example, in the field of food safety, the detection of odorants and flavor substances can significantly improve the quality control of raw materials, the supervision of processing process, and freshness degree [11,12,13,14,15,16,17]. In the field of safety and security, the detection of drugs and explosives can be achieved by the contact-free detection of characteristic odorants [18,19,20,21]. The assessment of air quality is very necessary to maintain and improve the quality of human settlements by detecting waste gas and harmful gases [22,23,24,25,26,27]. What’s more, the high performance on pre-detecting biomarkers of infection, poisoning or metabolic diseases can effectively improve the diagnostic efficiency of diseases [28,29,30,31,32,33]. In addition, the detection of odorants can contribute to monitoring quality and resisting counterfeit and abnormal ingredients in packaging materials, cosmetics and perfumes [34]. Notably, most of these odorants and tastants mentioned above can be detected by the ultra-sensitive human olfactory and gustatory systems. There are also professional sensory evaluators who specialize in the training of various odors and flavor. However, it is subjected to some limitations such as being time-consuming, subjective, unquantifiable and even dangerous [35]. Therefore, there is an urgent need for odorant and flavor detection technology to mimic the human olfactory and gustatory system in such fields.

With the development of science and technology, several researches have designed many detection strategies on odorants and tastants detection, such as patch clamp technique, gas chromatography-mass spectrometry (GC-MS), various sensors and so on. Patch clamp is the golden technique to detect the electrophysiological activity of cells [36,37], and gas chromatography-mass spectrometry (GC-MS) technique shows highly accuracy for odorant detection [16,38,39]. However, these techniques require extensive operation experience and related experimental instruments with high precision, which seriously hinder the wide practical applications for the detection of odorants. Although a portable gas chromatography-mass spectrometry device was developed for quick odor detection and analysis in real time, lower accuracy than laboratory equipment may be a great challenge for its development [40,41]. In various chemical sensors, sensitive materials greatly determine the overall sensing performance. Kinds of new materials have been applied for the development of sensors, for example, metal oxide semiconductor (MOS) sensors for odorant detection [42,43] and conductive polymer (CP) sensors for both odorant and flavor detection [44]. However, it is hard to detect odorants and tastants with these sensors similar to the biological olfactory and gustatory system [44,45]. Therefore, the biomimetic biosensor composed of bio-sensitive elements and transducer elements is a promising sensitive and effective detection strategy.

Mammals’ olfactory and gustatory systems have ultra-excellent characteristics of genetic diversity and a high performance in recognizing odorants and tastants, which can provide the most excellent sensitive materials for the construction of smell and taste sensory sensors. Many studies have successfully used animals to achieve high-sensitivity perception of the external chemical environment, such as well-trained canines for the detection of explosives [46,47,48,49,50]. However, animals require expensive maintenance and training/retraining costs, exhibit varies between individuals and species, and are highly susceptible to external interference [51]. Other rodents or insects, such as mice, bees and fruit flies, have also been proposed to be an alternative. Therefore, there is an urgent demand for a high-performance sensor system to realize the effective detection of dangerous chemicals with ultra-high sensitivity and high specificity, and stability. With the rapid development of modern sensor technology, it is possible to design the detection strategies based on olfactory and gustatory related biomolecule elements as the first sensing element, which can combine with different detection devices as the second sensing element [52,53,54]. Such sensors are called biomimetic olfactory biosensors or biomimetic gustatory biosensors. 

In the past decade, different biomimetic detection strategies have been developed, which exhibited many excellent characteristics, including high sensitivity and selectivity, good stability, and practicability, offering an effective platform for application in food safety [14,15,45,55], cosmetic fragrance, security [18,19,56,57], environmental security [22], auxiliary diagnosis and drug screening [31], and so on. The biomimetic detection system usually consists of four parts as shown in Figure 1. Various biomolecules related to olfactory and gustatory functions were used for the biorecognition, including membrane receptors, binding proteins, protein-derived peptides, nanovesicles, nanodiscs, related cells and even tissues used as sensitive materials (Figure 1A). In order to detect the interaction between sensitive materials and targets, various transducer devices with magnification function were employed as the second sensing elements of biomimetic sensors, such as field-effect transistor-based sensors, electrode-based sensors, and fluorescence-based sensors (Figure 1C). To realize the immobilization of sensitive materials on the sensing element, the deposition of sensing materials is also an important part of the detection strategy design. According to the structural properties of the sensitive materials and the surface physicochemical properties of the transducer, different immobilization designs have been applied, as shown in Figure 1B. Thereby, the principle of biomimetic detection strategies can be described as follows: (1) the related biorecognition molecules were successfully immobilized on transducer surface, (2) the target ligands were recognized selectively, resulting in the responsive signals of biorecognition molecules, and (3) the secondary transducer would collect and transduce the responsive signals into the form of voltage, current, impedance or optical changes. Notably, the mammalian olfactory and gustatory systems enable the recognition and detection of complex odorants and tastants with high sensitivity and efficiency due to their excellent encoding mechanism. The construction of biomimetic biosensor aims to mimic the mammalian olfactory/taste system as closely as possible. Therefore, except for the ability to detect odorants and tastants with high sensitivity, it also needs to perform the efficient encoding ability of the olfactory/gustatory system on complex ligands. To achieve this purpose, various pattern recognition algorithms were introduced in order to decipher responses and complete encoding function (Figure 1D), such as principal component analysis (PCA), support vector machine (SVM), and so on. The developed biomimetic detection strategies strive to retain and imitate the basic characteristics of the olfactory system and gustatory system, which also offer an effective research platform to understand the basic principle of biological perception.

This review will summarize the development of biomimetic olfactory and gustatory biosensors in the past 10 years and the possible direction of future research from the following four sections. The first section will introduce sensing mechanisms and principles of olfaction and gustation in mammals. The second section will introduce the immobilization strategies used for coupling between biomolecules and biomimetic sensory. The third section will elaborate various biomimetic sensors used for the detection of targets ligands with highlights. Finally, the current challenges and the possible direction of future research is discussed.

## 2. Sensing Mechanisms of Olfaction and Gustation

The working principles of biological olfaction and gustation are the important basis of biomimetic biosensing construction. Only by mastering and understanding their most basic working mechanism and principle can we simulate its basic working state as much as possible and develop the most effective, reliable biosensors in the process of utilization. Therefore, in this section, olfactory and gustatory sensing mechanisms are introduced as shown in Figure 2.

### 2.1. The Olfactory System

The olfactory system starts from the interactions between the odorant molecules and the olfactory receptors located on the cilia of the olfactory receptor neurons, which transduce chemical information of odorants into action potentials of neurons (Figure 2A). Then, the electrical signals are transmitted into the olfactory bulb and cortex for encoding, processing, and decoding. Finally, odorants are recognized and discriminated as shown in Figure 2A–C.

In olfactory receptor neurons, once the olfactory receptors bind with specific odorants, a cascade of cellular reactions will take place. As shown in Figure 2C, olfactory receptor belongs to the G protein coupled receptor (GPCR) family, the activation of the olfactory receptor will motivate G_αolf_-GTP and G_βγ_ formation. Then, the activated G_αolf_ protein will combine with adenyl cyclase III (AC III) to form a complex, which converts ATP into cyclic AMP (cAMP). Sustained increases of cAMP can trigger the opening of cyclic nucleotide-gated channels (CNGCs), resulting in the influx of Na^+^ and Ca^2+^ across the membrane. Simultaneous calcium influx opens calcium-activated chloride channels (CACCs) and allows chloride efflux across the membrane, eventually resulting in the “depolarization” on the membrane [53].

However, the olfactory receptors cannot be directly exposed to odorants in the air, but animals can still achieve highly sensitive odorant recognition. Several studies have found that odor-binding proteins (OBPs) may play an important role in the transport of odorants from air phase to liquid phase [58,59,60]. OBPs are small soluble proteins of the lipid carrier protein family with approximately 14 kDa and between 150 and 160 amino acids in length [61,62]. They are secreted by the olfactory mucus of the olfactory epithelium, and can reversibly bind volatile organic compounds. OBPs act as a transporter between the air-lymphatic interface and olfactory receptors and promote the interaction between odorants and olfactory receptors [63,64]. 

### 2.2. The Gustatory System 

Taste substances dissolve in saliva and active the taste buds in the mouth to stimulate the response of taste cells. These gustatory signals are transmitted to the high-level processing center through intracellular and extracellular signal transduction and afferent neurotransmission. After integrated coding, they are projected to the taste cortex of the cerebral cortex, and finally they produce taste sensation (Figure 2A,D).

The sensation of sweet, bitter, and umami depends on the specific recognition between taste receptors and tastants, which will activate the downstream G protein containing three subunits (G_α_, G_β_, and G_γ_). The guanosine diphosphate (GDP) on the Gα subunit is replaced by guanosine triphosphate (GTP). Then, the complex dissociates into G_α_-GTP and G_βγ_ subunits and activates the calcium ion channel, leading extracellular calcium influx and simultaneous endoplasmic reticulum releasing calcium (Figure 2E). Finally, the increase of intracellular calcium concentration leads to the potential difference between inside and outside the membrane [65]. 

Some molecular docking research showed that the combination of spicy substances and receptors [66] or binding proteins [67] involved spatial complementarity, hydrophobicity, hydrogen bonding, and π-π interactions. Therefore, the basic research of biological olfaction and gustation systems lay the solid foundation for the biomimetic biosensing construction.

## 3. Immobilization Strategies of Sensitive Materials

The immobilization of sensitive materials on the transducer surface is also an important aspect of the detection strategy design. For the construction of biomimetic-biosensors, it is required to maintain a certain bioactivity of various sensitive materials for effective discrimination and combination with target ligands. In addition, effective immobilization can realize the uniform adhesion or targeted distribution of sensitive materials on the transducer surface and help to achieve the effective, high-stability, and high-sensitivity detection of the target ligands. To fulfill these high requirements, different immobilization methods have been widely explored for different sensitive materials to improve the detection performance of biosensors. 

In the early stage, the drop coating is a simple and easy immobilization method, which is especially suitable for the immobilization of olfactory and gustatory-related protein molecules, cells and tissues [68,69]. However, this kind of physical immobilization strategy has many shortcomings, including difficulty ensuring the uniform and firm distribution of biomolecules on the transducer surface and difficulty guaranteeing the effect of the device substrate with low biocompatibility on the coupling degree of cell tissues on its surface. All of these problems will impact the stability, sensitivity and repeatability of the signal acquisition, and seriously hinder its further development. 

With the development of biotechnology and biomaterials, researchers have a more in-depth understanding of the physical and chemical properties and living environment characteristics of functional units related to olfaction and gustation. Chemical methods can achieve stable and efficient immobilization of sensitive materials. Olfaction and gustation related biomolecules can be modified with some functional groups for successful immobilization on the transducer surface through various chemical bonds, such as π-π stacking interactions between aromatics rings [55,70,71], Au-S bonds generated from thiol groups [72] and gold-based functionalized electrodes, nucleic acid sequence complementary pairing [73,74], etc. Meanwhile the specific modification of the transducer surface is also necessary to improve the efficiency of immobilization. For example, nanomaterials with a porous structure and abundant surface functional groups can significantly increases the specific surface area of the transducer, which can greatly enrich the immobilization strategy of sensitive materials on the transducer surface, such as carbon nanotubes (CNT) [75,76,77], reduced graphene oxide (rGO) [67] and gold nanoparticles [78,79].

For the cells or tissues, the external microenvironment seriously affects the healthy growth state and the normal function of cell tissues. In order to obtain the stable responses, it is quite crucial to simulate and maintain the normal growth state in vivo, which is the main consideration for the construction of cell and tissue-based sensing systems in vitro. The utilization of extracellular matrix on the substrate is an effective way [80]. And synthetic hydrogel are alternative choice, which is similar to the natural extracellular matrix and can be easily prepared in large quantities. What’s more, hydrogels can be formed through certain chemical crosslinking or physical crosslinking of all water soluble or hydrophilic polymers, such as polysaccharide (cellulose, alginic acid, hyaluronic acid, chitosan, etc.) and polypeptide (collagen, poly-L-lysine, etc.) [81,82]. Its easy preparation and diversity characteristics have greatly enriched the immobilization solutions on cell and tissue-based sensing systems in vitro.

## 4. Various Biomimetic Olfactory and Gustatory Biosensors

### 4.1. Biomimetic Olfactory and Gustatory Biosensors Based on Field-Effect Devices

Field-effect devices have been broadly used in the development of biomimetic olfactory and gustatory biosensors. In the field-effect devices, the conductivity of an underlying semiconductor layer can be modulated by the electric field of the gate electrode surface. It is an important physical mechanism of semiconductor devices. 

Table 1 summarized field-effect device-based biomimetic olfactory and gustatory biosensors for specific discrimination of odorants and tastants. In these systems, the sensitive biomolecules were immobilized onto the field-effect transistors, and the specific interaction between sensitive materials and target molecules will modulate the surface potential of gate electrode. Then, the signal responses can be detected and analyzed to discriminate the target molecules. Table 1 presents the basic parameters of biomimetic sensors and their potential applications.

#### 4.1.1. Field-Effect Transistor

Field-Effect Transistor (FET) is an important field-effect device with excellent characteristics including high input resistance, low noise, low power consumption, large dynamic range, and ease of integration. It consists of the source electrode, drain electrode, and gate electrode. When no voltage is applied to the gate, the current channel between the source and the drain is not turned on, and the FET is at the cut-off state. Once the applied voltage of the gate exceeds the threshold voltage, the current channel between the source and the drain will be turned on to form a leakage current. The leakage current can be tuned by the voltage of the gate with an internal amplification effect.

Recently, CNT as sensitive material carriers have been widely used to combine with FETs for the development of biomimetic biosensors, which can be attributed to: (1) the tubular structure providing abundant binding sites and a wider detection range, (2) the high electrical conductivity and large carrier mobility providing effective direction for improving the interfacial properties of transducer devices, and (3) the excellent chemical stability and elastic mechanical properties being beneficial to improving the specificity, selectivity, and stability of the sensor. In addition, many studies usually utilized equilibrium constant K to reflect the dissociation rate of the combination between proteins and small molecules so as to achieve quantitative evaluation of protein affinity level with small molecules. FET-based biosensors were developed by coupling olfactory receptors as sensitive materials to recognize different specific ligands, which can be widely applied in food quality analysis, disease diagnosis and environmental assessment. A single-walled carbon nanotubes-field effect transistor (swCNT-FET) was coated with polyethylene glycol (PEG) for the prevention of nonspecific absorption and functionalized with the human olfactory receptor 2AG1 (hOR2AG1), which was used to detect amyl butyrate (AB) selectively with ultrahigh sensitivity down to concentrations as low as 1 fM [83]. Carbon nanotube-based field-effect transistor (CNT-FET) was equipped with rectangular floating electrode functionalized with T1R2 venus flytrap (VFT) through N-acetyl-l-cysteine solution coating and amide bond formation, which achieved more ultrahigh detection on sweet tastants with a limit down to concentrations as low as 0.1 fM and highly stability with more than 28 days [84]. The dissociation constant of 10^–12^ M indicated the high affinity between VFT and sweet tastants and the similar constant of sucrose and sucralose contributed to their similar structure. 

Graphene has a large specific surface area, good stability, electrical conductivity and excellent absorbability, which has been widely used to modify and perfect the transducer surface. A bioelectronic tongue was established based on graphene-FET platform with VFT domain of human taste receptor type 1 member 1 (T1R1) as sensing elements and the disodium 5′-inosinate (IMP) as the enhancing element for the specific detection of umami taste in real time [85]. This platform not only shows high sensibility for capturing ligands but also has an effective reusable property and a great storage stability to 5 weeks. 

The developed biosensors have high resolution in distinguishing corresponding target molecules. Notably, the production of receptor proteins has the characteristics of simple, rapid and high-yield extraction, which provides a strong and beneficial basis for simulating human olfactory/gustation system. In Figure 3A, a portable and multiplexed biosensor was developed based on human olfactory and taste receptors with CNT-FET to achieve comprehensive information regarding food freshness from many apsects [85]. The system has a high *K_d_* value, the responsive range was calculated through Langmuir isotherm to be from 3.05 × 10^10^ M^−1^ to 7.53 × 10^12^ M^−1^, indicating the high affinity binding of the adopted receptors with different corresponding ligands. Such a detection strategy provides a new effective research direction for simulating human olfactory and gustatory perception. 

Although olfactory and gustatory receptors can simultaneously detect and recognize different ligands in real time, the processing and effective analysis of large amounts of data is time-consuming and error-prone. The combination of biosensor and machine learning has become the mainstream of current researches. Among these methods, principal component analysis (PCA) was especially suitable for dimensionality reduction of large quantities of data, effective and quick extraction of data features and rapid realization of visual research, which have been widely used in data analysis for distinguishing various smells and taste substances. As shown in Figure 3B, an artificial multiplexed super-bioelectronic nose (MSB-nose) was developed based on a liquid-ion gated FET coated with single layer graphene and functionalized with the assembly coding form of two different human olfactory receptors for the high-quality and high sensitive detection of AB and helional (HE) [86]. The PCA simulation results showed ultra-high accuracy with no less than 99%, indicating the MSB-nose can encode olfactory receptor combinations for distinct odor identification. Moreover, this biosensor successfully achieved super-highly sensitive AB and HE detection with a detection limit down to 0.1 fM and a good stability to 10 weeks. The equilibrium constant K of more than 10^14^ confirmed the perfect affinity between receptors and ligands.

In addition to receptors, OBPs are another effective alternative sensitive material with hydrophobic β-barrel cavity, successfully providing a useful model system to study binding affinity with odor compounds. Figure 3C showed a capacitive layer structure based on water-gated organic field-effect transistor and OBP immobilized on the gate, which successfully achieved sensitive and quantitative measurements of weak interactions associated with neutral enantiomers. The sensing system opened a new pathway for accurate derivation of conformational events related to OBP interactions with an ultra-weak molecule [87]. In another study, odorant-binding protein 14 (OBP14) from the honey bee was used as the sensitive material for quantitative recognition and detection of a variety of ligands containing a hydroxy group, such as homovanillic acid, eugenol, and methyl vanillate [78]. The results confirmed that OBP14 with a hydrophobic β-barrel cavity has a higher affinity with the ligands with the hydroxy group.

Although the solution to extract receptor or binding proteins may be very simple and fast, the conformation and biological activity of membrane proteins are still easily damaged during extraction due to various problems. In addition, it was still difficult to keep the activity, conformation maintenance and stability of extracted proteins in vitro. To challenge this problem, nanodiscs were reported and successfully extracted with a phospholipid bilayer structure composed of membrane scaffold proteins (MSPs) and phospholipids, providing powerful technical support for membrane protein research and olfactory and gustatory detection. Yang et al. developed a biomimetic olfactory biosensor based on oriented-immobilization nanodiscs embedded with trace-amine-associated receptor 13c (TAAR13c) as a sensitive element and CNT-FET with floating gold electrodes as a transducer element for highly-sensitive detection of death-associated odor, cadaverine (CV). The developed system successfully achieved detection on CV with a concentration range over five orders of magnitude and showed good selectivity toward other interference including diaminodecane, trimethylamine, ethanolamine, and glutamine. The equilibrium constant K between T13NDs and CV was estimated as 3.63 × 10^11^ M^–1^, showing the high affinity between ligands and receptors and different periods of spoilage experiments, indicating the excellent practical application value for evaluations in food safety environments [88]. To improve the signal responses with similar strategy, the team firstly proposed an auxiliary method with 1 nM benzyl salicylate as enhancer materials. Figure 3D shows the biomimetic olfactory biosensor based on oriented-immobilization nanodiscs embedded with human olfactory receptor 1A2 (hOR1A2), which has ultra-high sensitivity to geraniol with a detection limit down to 1 fM. The *K_d_* between hOR1A2 and geraniol was changing from 8.37 × 10^11^ M^−1^ to 1.64 × 10^15^ M^−1^ after the addition of enhancer materials, which indicated that the benzyl salicylate as the enhancer could obviously improve the binding affinity between ligands and receptors [89]. Recently, a portable biomimetic olfactory biosensor was successfully developed based on nanodiscs embedded with TAAR13c and TAAR13d for high detection of CV and putrescine (PT) with a limit down to 1 fM, respectively [15]. They also evaluated the binding degree of TAAR13 to CV and PT based on a SWISS-MODEL simulation [96], and the results showed that TAAR13 specifically combined with CV and PT was associated with Asp 112 and Asp 202 (both the amino groups of CV and PT formed salt bridges with Asp 112 and Asp 202 in TAAR13 at specific distances, respectively). Notably, the developed portable system can successfully achieve detection of diverse real samples on-site and monitor the time of freshness, providing a potential platform for the in situ and on-site monitoring of meat spoilage. However, utilization of nanodiscs embedded with receptors cannot completely analyze the ligand-induced intracellular signaling cascades. To challenge this problem, the emergence of nanovesicles containing both transmembrane proteins and contain cytoplasmic proteins, were successfully explored and widely applied. A biomimetic gustatory biosensor was proposed based on nanovesicles containing honeybee umami taste receptor, gustatory receptor 10 of Apis mellifera as sensitive materials CNT-FET with floating gold electrodes for high selectivity on l-monosodium glutamate (MSG) detection [90]. Notably, IMP has been widely used as the enhancing element for synergistic effects with MSG. Related reports suggested that the binding site of IMP close to human umami taste receptors expressing venus flytrap domain (VFTD) may strongly potentiate the umami taste intensity in insect taste systems, which was further confirmed with the equilibrium constant between AmGr10 and MSG changing from 1.77 × 10^8^ M^–1^ to 2.30 × 10^9^ M^–1^ in this developed biomimetic gustatory biosensor. Based on the similar detection strategies, another research proposed a new specific human olfactory receptor (OR) found using a cyclic adenosine monophosphate (cAMP) response element (CRE)-reporter gene assay and successfully achieved high sensitivity real-time detection of fungal contamination in grain down to 1 fM and high selectivity toward other interference including 1-octanol, 1-octanal, 3-octanol, and 1-octene [91]. Notably, some mammalian receptors composed of two different proteins, such as heterodimeric class sweet receptor based on hTAS1R2 and hTAS1R3 can also integrate with nanovesicles and achieve the detection and discrimination of sweeteners based on similar detection strategies, as shown in Figure 4A. Its estimated K values in real drink examples were all ∼2.0 × 10^–3^ M, showing that the developed sweet biosensor have great potential application in detection and monitoring sweeteners in real samples [75]. However, the complex cellular signaling processes induced by olfactory and taste receptors in response to specific ligands have always been difficult to analyze. To accurately analyze the GPCR reaction process, one research work proposed an effective method based on coupling ion channels with receptors to avoid complex cellular signaling processes and a successful reduction of the measurement errors caused by instability of various cellular components [92]. This work created an olfactory biosensor system in which nanovesicles expressing olfactory receptors were covalently fused to Kir6.2 channels, when specific ligands that altered the conformational structure of receptors would directly result in the influx of potassium ions to activate Kir6.2 channels. This nanovesicle-based bioelectronic tongue showed ultrahigh sensitivity on AB down to concentrations as low as 1 fM and a good stability to 8 days, which successfully offers an effective platform for GPCR reaction process analysis and an alternative to labor-intensive and time-consuming cell-based analysis and to widely sensory assessment in the food and beverage industry, while the similar detection strategies that achieved a different detection limit may be attributed to the affinity between receptors and ligands and immobilization strategies, such as π-stacking and charge-charge interaction through PDL function. According to the collected research, it can be found that the physical immobilization method may provide a more suitable and stable micro-environment for nanovesicles in response to specific ligands. Detergent micelles belong to amphiphilic phospholipid structures, which have been widely used for stable environment construction of recombinant GPCR, a bioelectronic nose based on detergent micelles as a recombinant environment for the nematode olfactory receptor, ODR-10 expressed in Escherichia coli (*E. coli*) [11]. This work successfully achieved highly sensitive detection of diacetyl with a limit down to 10 fM and distinguished it from other structurally similar substances in different environments, providing a useful tool for a variety of industrial applications.

Since an increasing number of micro-biomolecules was discovered and used as the sensing element, artificial synthetic peptide derived from receptors or the binding proteins’ specific binding sites can be synthesized as another substitution of protein due to its high yield and stability. Detection of salmonella infection is relatively difficult and meaningful. A bioelectronic nose was proposed based on CNT-FET functionalized with odorant binding protein (OBP)-derived peptide expressed in Drosophila for the rapid detection of salmonella contamination in ham [71]. The system successfully achieved the sensitive detection of 3-methyl-1-butanol with a detection limit down to 1 fM. The corresponding equilibrium constant was more than 10^13^, which suggested high affinity between peptides and targets. Unlike previous studies that used chromatographic instruments and strain counts to evaluate seafood freshness, a bioelectronic nose was developed based on CNT-FET functionalized with a synthetic peptide horp61m for rapid and non-invasive detection of gas trimethylamine (TMA). In order to evaluate the effectiveness of the biological electronic nose more accurately, the study adopted the sensory evaluation and gas chromatography-mass spectrometry (GC-MS) as a standard for comparison as shown in Figure 4B. The results showed that peptide-based olfactory biomimetic sensor can effectively simulate the human olfactory system, which was very significant for the validation of practical application value of constructing biomimetic biosensor system [55]. A bioelectronic nose was developed based on swCNT-FET combined with microfluidic system (µBN) and functionalized with synthetic peptide horp61m for highly sensitive and good selectivity on gaseous TMA detection depicted in Figure 4C. µBN as a device for precise control and manipulation of micro-scale fluids was applied and widely developed to simulate the transport process of odorant and tastant substances in the human body [70]. In particularly, the developed integrated system successfully achieved a high sensitivity of 10 ppt for gaseous TMA detection and distinguished gaseous TMA produced by real spoiled seafood (oysters) without any treatments from other types of spoiled food, showing great potential in real-time rapid field analysis and providing an effective direction for simulation of human olfactory and gustatory functions.

#### 4.1.2. Other Types of Biosensors Based on Field Effect Properties 

Electrolyte-insulator-semiconductor (EIS) sensors are another type of field effect device, which has been widely used in chemical and biological sensing applications due to its small size and simple fabrication method, as shown in Table 1. Utilization of EIS sensors as a transducer element can prosperously achieve detection on the function expression of sensitive materials through monitoring changes in transducer capacitance in respond to specific ligands. The biomimetic biosensors was fabricated based on EIS functionalized with ODR-10 [93], and 0.5% Brij58 was used as the detergent to promote the expression of the nematode olfactory receptor in *E. coli* in this work, providing an effective method for batch production of sensitive materials. The system has characteristics of miniaturization, stability, and easy access to sensitive materials, and it provides a potential development direction for portable olfactory or gustatory detection device construction.

LAPS are another type of field effect device with an EIS structure. As the transducer element, LAPS has excellent light-addressable ability, making it suitable for the signal capture of single cell and tissue in response to specific ligands, obviously except patch-clamp technology. In the development of biomimetic sensors, olfactory and gustatory epithelium were used as sensitive materials and LAPS as the detection device for the determination of specific ligand combinations through change of photocurrent acquisition [94,97]. A dual functional extracellular recording biosensor was developed based on taste bud cells cultured on the surface of LAPS functionalized with ATP-sensitive DNA aptamer to realize the simultaneous detection of the membrane potential and ATP release of single taste bud cells responding to specific bitter ligands [94]. Another interesting concept has been proposed based on electrolyte-semiconductor (ES) structures, especially the introduction of indium tin oxide (ITO) with good conductivity and transparency as a semiconductor, facilitating the sensing system more simply, at a lower-cost, and with sensitive detection in chemical and biosensing applications. The report developed a novel bioelectronic gustatory biosensor based on an ITO-ES structure equipped with *E. coli* expressing hT2R4 as a sensing element without any additional steps of protein extraction and cell/protein immobilization for successful bitter substances detection through monitoring of the variation of space charge capacitance [95].

### 4.2. Biomimetic Olfactory and Gustatory Biosensor Based on Electrode Array

#### 4.2.1. Micro-Electrode Array 

A microelectrode array (MEAs) system can record the electrophysiological signals of sensitive materials in vitro and in vivo [98]. Isolated cells or tissues can be cultured directly on the MEAs surface and the membrane potential will be changed due to membrane surface receptor combined with target ligands causing redistribution of charge, which can be detected by this system. The MEAs system possesses unique long-term monitoring electrophysiological characteristics, making it useful in many applications, such as high-throughput screening for medicine, detection of a combination of cell electrophysiological functions and morphology and so on. This part would introduce olfactory or gustatory biosensors based on MEAs recording system and different types of sensing elements, such as cells and issue, from two aspects of in vivo and in vitro culture, as shown in Table 2.

When the olfactory and gustatory systems are stimulated by external odorants and tastants, a cascade of reactions will occur in the corresponding cell and transport to respective intermediate stations for integration and procession. Direct utilization of in vivo mammalian system is a very effective strategy for olfactory and gustatory detection. An olfactory biosensor in vivo was proposed based on an array of mitral/tufted cells (M/Ts) as sensing elements coupled with MEAs as transducer elements for odorant discrimination [99]. The results showed the in vivo bioelectronic nose can successfully achieve natural odor detection and discrimination. Although the work indicated that a small percentage of M/Ts can carry enough information to distinguish odors, the position of the microelectrode in the olfactory bulb was not clear and the specificity and repeatability of the sensor could not be fully guaranteed. A genetically modified method was used as an effective solution to solve it. They developed murine M72 olfactory sensory neurons expressed with the green fluorescent protein through genetic modification for the relatively high precision location of multiple microelectrodes, which achieved high sensitivity and good stability and selectivity in long-term trinitrotoluene (TNT) detection [100]. This study demonstrated that specific responses to odorants can be regulated by adjusting the expression of specific ORs. In the latest study, researchers genetically engineered olfactory epithelium to overexpress specific OR genes based on serotype 9 recombinant adenovirus-associated virus (AAV) for the detection of specific odorants [101]. 

In addition, an in vivo gustatory system can also achieve effective detection for the high-quality detection of bitter compounds in food and medicine. In Figure 5A, a microelectrode array was coupled with a rat’s gustatory cortex for the selective and specific detection of a bitter compound down to 0.1 µM, exceeding both the in vitro conventional bioelectronic tongue and nose [102]. However, motion artifacts would generate when local field potentials were detected in the wake state of before and after bitter stimulation. Their work detected local field potentials in the rat’s anesthesia states and achieved high sensitivity detection and high specificity screening of bitter compounds [103].

Utilization of high sensitivity of the mammalian olfactory system directly can achieve supernal value in multi-type odor discrimination and can avoid the complex means of extracting cell tissues and proteins. However, the development of biomimetic biosensing systems based on in vivo animals requires researchers with excellent experimental skills and experience, laboratories with sophisticated equipment, as well as the difficult to solve electrode stability, which seriously hinders its extensive development and application. An effective alternative on balance to resolve this problem and simultaneously implement spatiotemporal analysis is to isolate the primary olfactory and gustatory epithelium and plate it on the surface of planar integrated MEA system. The extracted primary tissues or cells contain their biological characteristics without completely changing and simulating the process of odorant and tastant sensing in vivo to reflect the state of the body in a certain extent. What’s more, spatiotemporal methods involving time-domain and frequency-domain analysis were proposed and developed as the auxiliary analysis solution in such a sensing system to analyze the correlation between different channels and furtherly verify detection performance of the MEA system as the transducer element. In earlier work, [97] successfully obtained the olfactory code information of the intact olfactory epithelium induced by stimulating different ligands. Another bioelectronic taste sensor for natural and artificial sweeteners detection was developed based on the intact taste epithelium extracted and cultivated on the MEA surface as shown in Figure 5B [104]. In order to further visualize the response patterns of taste tissues to different sweet tastes, this work utilized another machine learning, k-means algorithm as an auxiliary analysis tool to differentiate the expression of different sweet taste responses. The k-means algorithm can realize the classification of clusters with different characteristics, which is very suitable for the classification of continuous data sets with small dimensions and values due to its advantages of easy implementation, simple principle, and fast clustering speed. In this work, the peak ordering diagram of all sweetener induced signals was analyzed through the k-means algorithm and the corresponding typical waveform was clustered to each stimulus, successfully realizing the differential response of taste tissues to different sweeteners. In addition, the algorithm analysis informed that different sweeteners had an obvious and different response on the detection system and the values of artificial sweeteners showed a low concentration that was bigger than that of natural sugars with a higher concentration. What’s more, the high specificity and detection effectiveness of this detection system was verified by its long-term signal recording results, dose-dependent increase curves obtained by the treatment with different concentrations, and mixture of the experimental results of two sweeteners with a similar chemical functional structure. The olfactory receptors contain metal binding sites and their conformational changes can be caused by binding to metals, such as colloidal nano-zinc isolated from blood, which might greatly improve the sensitivity of bioelectronics noses. One work confirmed the enhancer role of zinc nanoparticles and obtained the higher quality detection of electrophysiological signals produced by the olfactory epitheliums in response to different odors [68]. However, the limitation in stability and functional cellular localization of primary cell tissue extraction from the host animal always impedes such a biosensor’s development. To challenge these problems, primary cells as sensitive materials have widely developed and applied. One work developed the bioelectronic tongue based on the MEA system and the primary taste cells obtained from fungiform papillae in the frontal tongue of female adult Sprague Dawley rats for effective noninvasive electrophysiological recording of taste receptor cells after and before acid stimulation. For exactly confirming the function of ASICs and PKDL channels on the sour taste coding mechanism, they still utilized the patch clamp system for the self-construction of Hodgkin–Huxley type mathematical model of taste receptor cells with acid-sensing in the whole cell [105]. However, the coupling degree of cells to the electrode surface greatly affects the efficiency of signal transduction in the biosensor system. Reducing the distance between the cells and the electrode surface to reduce the sealing resistance is an alternative effective solution. Specific hybridization between short complementary ssDNA strands was utilized to achieve controllable orientation of olfactory cell immobilization sites on the sensor surface, which significantly improved cell signal transduction efficiency as shown in Figure 6A [106]. In addition to olfactory and gustatory related cells, olfactory and gustatory receptors can also be expressed in other systems, such as the gastrointestinal and respiratory tracts of mammals, the male reproductive system, and the brain and heart, providing a variety of possibilities olfactory and gustatory biomimetic biosensor construction. In recent research, primary cardiomyocytes endogenously expressing bitter taste receptors (Tas2r) and umami taste receptors (Tas1r1 and Tas1r3) were utilized as sensitive materials combined with the MEA system for the detection of bitter and umami substances, as shown in Figure 6B [82].

#### 4.2.2. Electric Cell-Substrate Impedance Sensing

Electric Cell-substrate Impedance Sensing (ECIS) can simultaneously measure the resistance changes and membrane capacitance changes of multiple groups of cells, as well as cultured cells and basement membranes, which was also summarized in Table 2. Activation of specific receptors on the cell surface will cause changes in cell morphology. Through the measurement of impedance spectrum, the ECIS technique can obtain and analyze the relationship between cell morphological changes and stimulation, extracellular matrix and cell proliferation in real time, providing abundant data for olfactory and taste signal transduction and coding and decoding processing. Except for cells specialized in endogenous expression gustatory receptor, several special cells with other functions also express some specific taste receptor, such as male mouse germ cells [107,108]. In Figure 6D, the bioelectronic taste sensor was developed based on ECIS functionalized with germ cells, expressing bitter receptor T2Rs for the specific detection of a bitter compound [108]. In addition to the function of combination and identification of bitter compounds, several agonists of specific bitter receptors have been found to perform great anticancer effects, such as N-C=S-containing compounds, allyl-isothiocyanates [107]. As shown in Figure 6C, the bioelectronic taste sensor was developed based on ECIS functionalized with human Caco-2 cells endogenously expressing T2R38 for the high-specificity detection of isothiocyanate-induced bitter compound. Moreover, the ligand-based virtual screening protocol was applied to filtrate ligands that are likely to activate the T2R38 receptor, and calcium ion imaging and inhibition experiments were performed to furtherly ensure the feasibility of cell-based BioET in this study. 

**Table 2 biosensors-12-00858-t002:** Summary of detection strategies on biomimetic olfactory and gustatory biosensors based on electrode array.

Detection Methods	Sensitive Material Types	Target Compound(s)	Detection Limit or Minimum Detection/Detection Range	Selectivity	Data Analysis Auxiliary Algorithm	Potential Applications Fields	Ref.
MEAs in vivo for electrophysiological detection	The mitral/tufted cells	The natural stimuli (banana, orange, pineapple, strawberry) and the monomolecular odors (isoamyl acetate, citral)	-	-	PCA (isoamyl acetate and banana; orange and citral) (PC1: 64.3%, PC2: 20.6%, PC3: 6.6%) (strawberry, pineapple, orange, banana) (PC1: 67.7%, PC2: 14.4%, PC3: 10.7%) for different ligands classification	Security Application	[99]
	Labeled murine M72 olfactory sensory neurons with the green fluorescent protein	Trinitrotoluene	10^−5^ M/10^−6^ M to 10^−3^ M	Acetophenone and methyl salicylate	-	Security Application	[100]
	The mitral/tufted cells	Nonanoic acid, octanoic acid, amyl hexanoate, and lyral	10^−5^ M/10^−6^ M to 10^−3^ M	Heptanoic acid, amyl hexanoate and ethyl acetate	PCA (PC1: 60.6%, PC2: 26.6%, PC3: 6.9%)	Food application	[101]
	The gustatory cortex	Denatonium benzoate	0.076 μM/10 μM to 100 mM	Sucrose, NaCl, HCl	PCA (PC1: 31.7%; PC2: 27.6%) for different ligands classification	Drug screening	[102]
		Quinine HCl denatonium benzoate salicin	The typical bitter-responsive neuron (BRN): (8.95 μM, 52.0 μM, and 14.1 μM, respectively.) and bitter-specific LFP envelopes (56.3 μM, 0.130 μM and 21.4 μM, respectively) / 1 mM to 100 mM	Sucrose, NaCl, HCl	PCA (PC1: 45.8%; PC2: 37.4%). SVM: 94.05% for different ligands classification		[103]
MEAs in vitro for electrophysiological detection	The taste epithelium	Isoamyl acetate and acetic acid	-	-	K-means algorithm	Food additives	[68]
	Primary cardiomyocytes	Denatonium Benzoate, diphenidol, monosodium glutamate	3.46 × 10^−6^ M, 2.92 × 10^−6^ M, and 1.61 × 10^−6^ M, respectively/10 μM, to 640 μM, 5 μM–320 μM, and 1 μM to 4000 μM, respectively	Sucrose, NaCl, HCl	PCA (the cumulative contribution rate is 94.9%)	Drug screening	[82]
	Primary taste cells obtained from fungiform papillae	Glucose, sucrose, saccharin, cyclamate	Glucose (50 mM–150 mM) and saccharin (5 mM to 15 mM)	-	K-means algorithm	Food application	[104]
	Olfactory cells prepared from the olfactory epithelium	The hydrochloric acid solutions	-	-	Whole-cell Hodgkin–Huxley type mathematical model		[105]
ECIS for impedance	Male mouse germ cells expressing T2Rs	Octanal and hexanal	0.1 mM	-	-	Flavoring agent	[106]
	The Caco-2 cell line endogenously expressing T2R38	Denatonium, nphenylthiourea, 6-propyl-2 thiouracil, quinine	78 nM, 4 nM, 0.4 nM and 62.5 nM, respectively/83 μM to 5000 μM, 4 μM to 200 μM, 0.5 μM to 50 μM and 62.5 μM to 4000 μM, respectively	Glucose, monosodium glutamate, NaCl, HCl	-	Drug screening	[108]
		Phenylthiocarbamide and propylthiouracil	0.09352 μM and 0.8404 μM, respectively/1 μM to 1 mM	Glucose, monosodium glutamate, NaCl, HCl	The ligand-based virtual screening protocol (Screening agonists: thiosinamine, α-Naphthylthiourea, and phenyl isothiocyanate)	Drug screening	[107]

### 4.3. Biomimetic Olfactory and Gustatory Biosensor Based on Electrochemical Detection Method

The electrochemical analysis method has been widely used in application of biomimetic olfactory and gustatory biosensors due to its simple instrument, high sensitivity, high detection efficiency, and accuracy, as shown in Table 3. The charge changes between the electronic conductor and the ionic conductor were detected to realize the qualitative and quantitative analysis of the substances according to the electrochemical properties of the substances in the solution and their change law. The three electrode systems consist of the working electrode (WE), reference electrode (RE), and auxiliary electrode (AE). The working electrodes were modified with packaged sensitive materials, the response of specific substance simulation was detected by monitoring the changes in its morphology, conformation, and electrical conductivity.

Glassy carbon electrodes (GCE) are commonly used as working electrodes in biomimetic biosensor construction due to its excellent characteristics of good electrical conductivity, high chemical stability, and small thermal expansion coefficient [109,110,111]. The combination of receptors and ligand has gradually become a popular issue in the bio-sensing field. The activation constant (*Ka*) was usually used to evaluate the affinity of receptors with ligands. The smaller the *Ka* value is, the higher the signal output efficiency produced by the ligand-receptor interaction. In addition, molecular docking can search the best matching pattern between molecules and ligands through geometric matching and energy matching mutual recognition, providing an effective analysis platform for exploring the combination pattern and affinity of olfactory and gustatory receptors and ligands. Figure 7A illustrated that a double-layer gold nanoparticle (AuNPs) biosensor was developed based on GCE and human umami receptor hT1R1 toward four umami ligands detection (MSG, IMP, disodium guanylate (GMP), and disodium succinate (SUC)). The calculated *Ka* through the sensing kinetics method showed that GMP have the highest affinity combined with hT1R1. Additionally, the analysis results of molecular docking model showed that the tight binding of TIR1 to the ligands depends on the formation of hydrogen bonds between its two rings (Asp147 and Ala170) and the amino/imino groups of the ligands. Notably, the hydrogen bond also formed between the phosphate group in GMP and IMP and the amino group at Asn69 to render the VFT active bag more tightly with ligands, which explained the reason why IMP and GMP can perform as enhancers for promoting umami perception [66]. What’s more, the team also utilized the double-layer gold nanoparticle biosensor and combined with sensory evaluation as a control to realize the detection of five umami peptides produced from *T. flavidus*. The quantum chemistry and molecular docking results demonstrated that aspartic acid was the active site in the umami peptides. The meaningful work revealed the umami mechanism of peptides and provided a reasonable tool for rapid screening of umami peptides in food. Another interesting research, as shown in Figure 7B, directly utilized VFT as the sensitive material element and designed a dual-signal amplifying sensor via CS-modified SWCNTs and PB-modified AuNPs-T1R1-VFT, which achieved highly sensitive detection on umami substances with a limit ranging from 10 fM to 100 fM and good selectivity toward other interference [112]. However, receptor purification steps are unavoidable in exploring the mechanism of ligand-receptor interaction of receptors in vitro. When the receptor was extracted from the cell membrane, the docking interaction with the membrane was lost. The conformational change caused by the ligand cannot fully achieve its conformational morphological change on the membrane. To challenge this problem, one research utilized a receptor protein embedding membrane as the sensitive material element to maintain its microenvironment and conformational morphology in respond to ligands as shown in Figure 8A. Its calculated Ka down to 10^−15^ showed capsaicine has the highest affinity with hTRPV1. Molecular docking simulation analysis demonstrated that Glu570 in the active pocket of hTRPV1 has a vital function in identifying spicy substances [67]. Furthermore, it also indicated that the combination of spicy substances and receptors can be promoted by spatial complementarity, hydrophobicity, hydrogen bonding and π-π interactions together. The extracted primary tissues can reflect the real state in vivo to some extent, which provides an effective tool for studying the interaction mechanism between receptors and ligands in vitro. However, it may be difficult for the cell/tissue with a certain weight to functionalize the glassy carbon electrode. To increase the amount of taste bud tissue attached to the biosensor surface and to prevent the tissue from flowing or falling off, a sandwich-type format was designed, which is composed of aldehyde starch gel solution and sodium alginate solution in proportion to form an upper and lower interlayer, and the taste tissue was successfully immobilized on the surface of the working electrode as a core. 

Recently, taste buds were extracted from different regions of rabbit tongue tissue to construct a biomimetic biosensor, and its auxiliary analysis indicator *Ka* showed that the umami receptors in different tongue regions have different perceptions of MSG and IMP as shown in Figure 7C [65]. Another biosensor based on primary taste bud tissue and GCE can successfully measure capsaicin with a calculated activation constant reaching 2.0218 × 10^−18^ M and also analgesic compounds up to the attomole-level. Its related competition experiments and quantitatively the antagonistic strengths and inhibition constants investigation showed that capsaicin, AMG 517, lurelin B and tetrahydropalmatine are competitive allosteric regulatory ligands of capsaicin, while aconitine and anandamide are non-competitive and competitive mixed allosteric regulatory ligands. Results of these studies showed that such primary taste bud biosensors can precisely mimic the ligand-receptor interaction environment in biological taste systems to quantitatively illustrate the dynamic characteristics of cell or tissue receptor interaction with its ligand, the effect of cell signal cascade, and the amounts of receptors and signal transmission pathways. However, GCE is easily polluted by some organic and metal compounds. Screen printing electrode also has been widely used in environmental security detection, auxiliary clinical diagnosis, and food composition detection [113,114]. It is a disposable electrode with advantages of being low-cost, maintenance-free, and multifunctional, with high repeatability between electrodes. Because of its high electrochemical performance, increasing numbers of studies have utilized it as the second transducer and made it functionalized with different types of receptors or cells to build up biomimetic biosensors for the detection and discrimination of different ligands. A novel taste biosensor was developed based on a screen printed carbon electrode (SPCE) functionalized with human recombinant OBP II a for detection of fat taste substances with a different length of alkyl chain as shown in Figure 8B [115]. For the detection and discrimination of complicated mixture components, two types of biological tongues were developed based on NCI-H716 from human enteroendocrine and STC-1 cells from mouse enteroendocrine, respectively [113]. The results demonstrated that STC-1 cell-based SPCE biosensor was highly correlated with quinine concentration and its taste perception ability was much better than that of the NCI-H716 cell sensor.

The interdigitated electrode (IDE), another alternative type of electrode, is a finger-shaped or comb-shaped electrode with periodic patterns in the surface obtained by electrochemical processing. As the core component of electrical signal transmission, it has been widely utilized in various applications, such as biomedical, environmental monitoring, and food safety due to its high collection rate, high signal-to-noise ratio and feedback effect similar to amplification. When the receptor immobilized on the IDE electrode surface binds to the corresponding ligand, its conformational change might increase the contact area between the mobile ion and the electrode, thereby making the electron transfer process easier, which has a great influence on the dielectric and conductive properties of the staggered electrodes. In Figure 8C, an olfactory biosensor was developed using IDE as transducer element and OBPs from an oriental fruit fly as sensitive material for the detection of insect semiochemicals at low concentrations through surface impedance variation monitoring [116]. The molecular docking model showed that the three-dimensional structure of BdorOBP2 has a central hydrophobic pocket, exposing a variety of chemical groups connected with pheromone chemicals through hydrophobic, polar, electrostatic, and π-stacking interactions to form stable conformational changes. Olfactory receptor-derived peptides (ORPs) were chemically immobilized onto single-walled carbon nanotubes (SWCNTs) on interdigitated electrodes based on Steglich esterification reaction (SER) and native chemical ligation (NCL), which achieved high responsiveness and stability to TMA with a detection of limit down to 0.01 ppt [117]. This novel ORP sensor has potential applications for environmental safety, food quality control, and healthcare.

**Table 3 biosensors-12-00858-t003:** Summary of detection strategies on biomimetic olfactory and gustatory biosensors based on electrochemical detection methods.

Detection Methods	Sensitive Material Types	Target Compound(s)	Detection Limit or Minimum Detection/Detection Range	Stability/Repeatability (RSD)/Reproducibility (RSD)	Ligand and Receptor Binding Index	Potential Applications Fields	Ref.
GCE as the working electrode	hT1R1	Sodium glutamate, disodium inosinate, disodium guanylate, and disodium succinate	1.5 pM, 0.88 pM, 2.3 pM and 0.86 pM. respectively/10^−14^ to 10^−12^ M	-	*Ka*: 7.420 × 10^−16^ M, 1.899 × 10^−15^ M, 5.449 × 10^−16^ M and, 2.251 × 10^−15^ M, respectively	Food application	[66]
	hT1R1	Five umami peptides: DVILPVPAF, TVAGGAWTYNTTSAVTVK, AMLEQVAMTDK, IGAEVYHNLK and GGKLVVDGHAIT	0.258 mM, 0.799 mM, 0.273 mM, 0.858 mM, 0.437 mM, respectively./DF9, TK18 and AK11: 10^−13^ g/L~10^−11^ g/L; IK10 and GT12: 10^−12^ g/L to 10^−10^ g/L.	-	*Ka*: 6.85 × 10^−13^ M^−1^, 7.40 × 10^−13^ M^−1^, 7.06 × 10^−13^ M^−1^, 6.68 × 10^−12^ M^−1^, 1.12 × 10^−11^ M^−1^, respectively.	Food application	[118]
	The venus flytrap (VFT) domain of T1R1	Inosine-5′-monophosphate, monosodium L-glutamate, beefy meaty peptide, and sodium succinate	0.1 pM, 0.1 pM, 0.1 pM, and 0.01 pM, respectively. And 10^−13^ M to 10^−6^ M, 10^−13^ M to 10^−8^ M, 10^−13^ M to 10^−7^ M, and 10^−14^ M to 10^−8^ M, respectively.	4 days (90%)/-/2.3–3.2%	-	Food application	[112]
	The cell membrane contained TRPV1	Capsaicin, allicin and sanshool	1 × 10^−15^ M, 1 × 10^−14^ M and 1 × 10^−15^ M, respectively./1 × 10^−15^ to 1 × 10^−13^ M, 1 × 10^−14^ to 1 × 10^−12^ and 1 × 10^−15^ to 1 × 10^−12^ M, respectively.	6 days (76.39%)/2.72%/0.75%	*Ka*: 3.5206 × 10^−16^ M, 5.0227 × 10^−15^ M and 1.7832 × 10^−15^ M, respectively.	Food additives	[67]
	The rabbit tongue bud tissue	L-monosodium glutamate and disodium 5′-inosinate	10^−13^ to 10^−8^ M, 10^−13^ to 10^−6^ M, respectively.	More than 5 days (81.75%)	*Ka*: 2.811 × 10^−14^ M^−1^ to 3.620 × 10^−12^ M^−1^, 1.716 × 10^−14^ M^−1^ to 2.526 × 10^−14^ M^−1^, respectively.	Food application	[65]
	The SD rat taste bud tissue expressing TRPV1	Capsaicin and mixed with analgesic compounds (capsazepine, AMG 517, loureirin B, aconitine, anandamide and tetrahydropalmatine)	1 × 10^−18^ M, -/1 × 10^−18^ M to 1 × 10^−15^ M	-	*Ka*: 2.0212 × 10^−18^ M, 7.159 M, 11.7241 M, 14.5442 M, 2.4218 M, and 2.4689 M, respectively.	Food additives	[111]
	The SD rats taste-bud tissues	Capsaicin, gingerol	1 × 10^−13^ M and 1 × 10^−12^ M, respectively/10^−13^ M to 9 × 10^−13^ M and 1 × 10^−12^ M to 3 × 10^−11^ M, respectively	More than 3 days (94.3%)/7.34%/9.03%	*Ka*: 1.249 × 10^−12^ M and 1.564 × 10^−11^ M, respectively.	Food additives	[109]
	The porcine taste bud tissues	Sucrose octaacetate, denatonium benzoate, quercetin	1 × 10^−14^ M, 1 × 10^−13^ M, 1 × 10^−14^ M, respectively/10^−14^ to 10^−12^ M, 10^−13^ to 10^−8^ M and 10^−14^ to 10^−9^ M, respectively	More than 7 days (90.09%)/RSD: 5.34%/RSD (Δ I): 1.72%	*Ka*: 8.748 × 10^−15^ M, 1.429 × 10^−12^ M, 6.613 × 10^−14^ M, respectively.	Food application	[110]
	The porcine taste bud tissues	L-glutamate	10^−16^ M/10^−16^ M to 10^−13^ M	-/RSD: 7.00%/RSD: 7.71%	*Ka*: 1.84 × 10^−16^–3.63 × 10^−15^	Food application	[119]
Gold disk electrodes for EIS detection	Liposomes containing Or10a, Or22a and Or71a	Methyl salicylate, methyl hexanoate and 4-ethylguaiacol, respectively.	1 pM, 1 fM and 0.1 fM, respectively./10^–13^ M–10^−7^ M, 10^–15^ M to 10^−7^ M and 10^–17^ to 10^−9^ M, respectively.	-	-	Cosmetics and medicine	[34]
SPCE as the working electrode	OBP II	Docosahexaenoic acid, linoleic acid, lauric acid	6 × 10^−11^ mg/mL, 10^−10^ mg/mL and 10^−9^ mg/mL, respectively./10^−8^ mg/mL to 10^−4^ mg/mL, 10^−9^ mg/mL to 10^−4^ mg/mL and 10^−9^ mg/mL to 10^−4^ mg/mL, respectively.	-	The dissociation constant: 10^−6^ M, 10^−9^ M and 10^−6^ M, respectively	Drug screening	[115]
	The taste receptor cells (NCI-H716 and STC-1 cells)	The sweetener mixtures and tastant mixtures.	-	-	-	Food application	[113]
IDE for the impedance variation detection	OBPs from oriental fruit fly	Isoamyl acetate, β-ionone, and benzaldehyde	3.3 × 10^−8^ M, 6.2 × 10^−8^ M, and 8.4 × 10^−8^ M, respectively/10^−7^ M to 10^−4^ M	2 weeks/-	-	Flavoring agent	[116]
	The olfactory receptor-derived peptides (NQLSNLSFSDLC)	Trimethylamine	0.01 ppt/0.1 ppb to 10 ppb	-	-	The spoiled food	[117]

### 4.4. Olfactory and Gustatory Detection Strategies Based on Optical Measurement

#### 4.4.1. Ca^2+^ Imaging

Intracellular calcium ions have temporal and spatial properties in neurocyte communication. In the resting state, the calcium ion concentration in most neuronal cells is approximately 50 to 100 nM. Once the cell is excited, intracellular calcium ion concentrations will instantaneously rise to 10 to 100 times. Therefore, detection of the variation of intracellular calcium ion concentration could be an effective way to monitor the cell mobility through special fluorescent dye or a calcium ion indicator. The influx of extracellular calcium ions includes voltage-gated calcium ion channels, ionic glutamine receptors, and nicotinic cholinergic receptors. The olfactory or taste biosensors introduced here were mainly based on calcium influx of ligand-gated channels as shown in Table 4.

Calcium ion indicators include chemical calcium ion indicators and genetically encoded calcium ion indicators. Fluo-4 AM was used as a chemical calcium ion indicator in an olfactory biosensor in which HEK-293 cells expressing hORs was cultured on the porous membrane to construct the microfluidic device for odor detection [81]. As receptors on the membrane surface bind to specific odor molecules, the calcium concentration greatly increased from the opening of the calcium ion channel and the release of endoplasmic reticulum into the cytoplasm. Then, the intracellular Fluo 4 would combine with calcium ions and emit green fluorescence under a 488 nm excitation light, which can be captured for odor information analysis. A sperm-cell gustatory biosensor was developed based on this Ca^2+^ imaging principle and innovative combination with flow cytometry to capture the green fluorescence for successfully bitter substance detection (Figure 9C) [122].

A series of cellular cascades and signaling codes constitute the overall response to external stimuli. To simulate the signal transduction environment of cells in vivo, the idea of a co-culture system was proposed. It has been widely used in many studies, such as oncology [126] and tissue engineering [73] to clarify intrinsic information transmission and material transportation in biological tissue [73,74]. A co-culture in vitro system was constructed based on gustatory cells and neurons to monitor cellular signaling and substance exchange and to better understand the mechanisms of taste perception [73]. This system made gustatory cells and neurons incubating with synthesized DNA-lipid conjugates, which was composed of single-stranded oligonucleotides, polyethylene glycol, and phospholipids, in order to achieve the close connection between the two kinds of cells. When the specific bitter molecules stimulate a co-culture in vitro system, continuous calcium influx responses between gustatory cells and neurons can be monitored by Fluo 4-AM. However, all neurons are also stimulated by bitter substances, which may cause more calcium ion influx and increase the complexity of data analysis of this system. For improvement of selectivity of calcium influx [73], agarose gel was used to cover the surface of the co-culture in vitro system to reduce the influx of calcium ions in nerve cells stimulated by taste substances as shown in Figure 9B. Results showed the feasibility of this approach, which can be used to improve the signal-selective cell response of multiple types of cells in a co-culture in vitro system [74].

Genetically encoded calcium ion indicators were another effective alternative to detect specific ligand analysis for the development of the olfactory biosensor. In Figure 9A, the GCaMP3-expressed fruit fly’s antenna is composed of a calcium ion binding calmodulin domain and a green fluorescent protein [120]. The conformation and fluorescence properties of GCaMP3 combined with calcium ion will be changed. This study also utilized multidimensional analysis for the discrimination of different types of cancer. Other analogical report constructed the microfluidic device based on Sf21 cells derived from *S. frugiperda* transfected with odorant receptor genes and GCaMP3 genes for long-term odor detection [121].

#### 4.4.2. SPR Measurement

Surface plasmon resonance (SPR) is a plasma wave-based optical measurement method, which has the excellent characteristics of real-time, label-free, and high-throughput monitoring for target detection. The signal is proportional to the refractive index (RI) of the medium near the gold metal film, indicating that any changes in the medium environment at a distance of 300 nm from the surface can be captured, which can perform an effective highly-sensitive transducer element for biomimetic olfactory and gustatory biosensor construction. When sensitive materials combine with specific ligands on the SPR surface, the changes of properties of the surface medium or the amount of attachment would cause resonance angle variation. A biomimetic olfactory biosensor was functionalized with liposome containing hOR3A1 as the sensitive material and based on SPR as the transducer element for the detection of helional [125]. Notably, in addition to functional peptides from olfactory binding proteins or receptor protein-binding domains, peptide sequences can also be obtained from antibody sequences of target receptors in virtue of molecular docking tools. Three TNT-binding peptide sequences were successfully obtained from the CDR1 heavy chain of anti-TNT antibody through antigen docking simulation. The function of the peptide was evaluated by SPR [56]. SWCNT was used as the transducer substrate to enlarge specific surface area through π stack functionalization for significant improvement of sensitivity and selectivity with the detection limit changing from 3.4 ppm to 772 ppb on 2,4,6-trinitrotoluene (TNT) recognition [18]. The system showed excellent performance on TNT explosives with enhanced sensitivity and excellent selectivity and long-term stability, providing an effective platform for environmental safety applications. 

High-throughput analysis has always been an important work of mass component detection in life science research. Surface plasmon resonance imaging (SPRi) offered a very promising multi-functional high-throughput research platform for complex odorant and tastant recognition. SPRi can be developed based on microarray device, which has all the advantages of SPR and can successfully realize real-time monitoring of various molecular action processes. A biomimetic olfactory biosensor was successfully developed based on SPRi as the transducer element for β-ionone and hexanal detection [63]. In this work, the SPRi combined with OBP derived from OBP3 by modifying different amino acid residues, and the VOC-induced conformational changes of OBPs were detected based on the SPRi signals amplification. Different receptor materials were functionalized in the microarray, indicating that the obtained olfactory/taste encoding information is nonlinear, abundant, and complex compared with that composed of pure receptors. With the continuous maturation of the extraction technology of various olfactory/taste receptors, olfactory binding proteins and their derived peptides, and the development of the combined analysis ideas with various popular algorithms, development of such sensors will become mainstream in biomimetic olfactory and gustatory biosensing platform construction. The SPRi microarray detection greatly provides a potential platform for mixture of odorant and tastant detection and furtherly promote development of biomimetic olfactory and gustatory biosensors to replace mammals smell/taste function in real life.

#### 4.4.3. Other Types of Optical Measurement Methods

The luciferase reporter gene system is another efficient fluorescence measurement method which uses luciferin as a substrate to detect firefly luciferase activity. A bioluminescence system consists of luciferin and luciferase, which is an effective detection method for detecting the interactions between transcription factors and DNA in the promoter region of the target gene. It was used to develop a dataset for other researchers to analyze olfactory coding information [127]. The in-vivo cultured cells could express the same specific OR and respond to specific odor, which can be monitored through SynaptopHluorin imaging [7]. 

In addition, interferometer imaging can realize the combination of one or more light waves and produce interference patterns. The interference patterns produced by different light wave combinations produced by different types of receptor-ligand binding on the working elements may be different. The contained olfactory and taste coding information may be very abundant and worth analyzing. Aryballe Corporation developed a novel portable olfactory biosensor based on Michelson interferometer functionalized with different types of functional peptides and equipped with machine learning as the fast analysis method, which has been commercialized and widely used in the automotive industry [128].

### 4.5. Biomimetic Olfactory and Gustatory Biosensor Based on Mass Sensitive Device 

Quartz crystal microbalances (QCM) is a typical quality sensitive sensor based on the piezoelectric effect of quartz crystal. With its high sensitivity, good chemical stability, and quantitative detection of high precision, it has been widely applied in many fields of biosensor technology, such as detection of the interactions between biological macromolecules, gaseous material analysis, etc. The interactions between the biomolecule and the target ligand can be monitored by detecting the output of the variation frequency. Table 4 summarized related biosensors. The olfactory receptor proteins from bullfrogs were coated on the surface of piezoelectric electrode for the detection of resonant frequency changes caused by the binding with specific ligands [69]. Aptamers facilitated the immobilization of olfactory receptor of *C. elegances*, ODR-10 onto QCM for effective detection of diacetyl [123]. The bitter receptor, T2R4, was immobilized onto the gold electrode surface for highly sensitive detection toward denatonium down to 5 nM (Figure 9D) [124]. The designed protein-mimetic peptide (PMP) motif from HarmOBP7 was immobilized onto QCM for long-chain aldehyde substance detection as shown in Figure 9E [72,129]. The excellent characteristic of easy synthesis, easy modification, high yield, and stability of peptides make it widely developed as a sensitive material for various biomimetic biosensors. Molecular docking technology can not only be used to explore the binding mode and affinity of receptor ligands but also to provide a suitable platform for the analysis and screening of new peptides through introducing predictive models to optimize trial-and-error analysis protocols and to minimize experimental issues such as non-specific identification. It was found that the amino acids in the composition of the peptide determine the affinity of the peptide to various olfactory and gustatory molecules, such as effective aspartic acid, is the active site of the umami peptide to capture ligands. Another research performed a virtual screening analysis to evaluate the affinity binding properties of 5 different peptides (cysteinylglycine, glutathione, Cys-Ile-His-Asn-Pro, Cys-Ile-Gln-Pro-Val, Cys-Arg-Gln-Val-Phe) with 14 volatile compounds simultaneously compared with the detection results of the biosensor constructed based on QCM as shown in Figure 9F. The experimental results showed that virtual screening can predict the sensing ability of pentapeptides with an accuracy rate of 93% [79]. In a follow-up work, the team used a semi-combinatorial virtual approach to extract a subset of 120 tripeptides from the complete tripeptide library (8000 elements) that were used as scaffolds to generate 7912 tetrapeptide combinatorial libraries. Then, according to the experimental procedure, five tetrapeptides (IHRI, KSDS, LGFD, TGKF and WHVS) were selected via virtual affinity and cross-reactivity. Together with PCA analysis, the results demonstrated that the final running virtual screening equipped with experimental conditions required to implement the biosensor closely matched with the real experimental data obtained from the biosensor array [78]. Therefore, the virtual screening method provides a new potential tool for peptide profile expansion and offers a fast, low-cost, and non-invasive procedure for peptide-based gas biosensors to distinguish different targets.

## 5. Future Prospects

The high-detection performance pursuit of biomimetic biosensors used for olfactory and gustatory molecular detection has drawn great attention in academic research. Based on existing research strategies, different detection strategies are summarized in the Table 5. These detection strategies have their own advantages and disadvantages and still have some room for improvement. The outstanding performance of bionic biosensors has been pursued including the lowest detection limit, ultra-selectivity and ultra-sensitivity, higher stability and repeatability, the greatest utilization valuation and economic benefit, especially high-throughput determination ability. Notably, factors contributing to the high-detection performance of biomimetic biosensors can form the following perspectives: (1) an enhancer used to improve the combination between biomolecules and taste and odor molecules, (2) a biomolecule used as a sensitive material, and (3) a detection method used to detect signals.

### 5.1. Enhanced Elements

The enhanced elements were added into the medium to improve the responses of sensitive materials to specific ligands via the allosteric modulation mechanism. For example, zinc nanoparticles located in the interface between the guanine nucleotide binding protein and the receptor proteins to improve signal transmission when responding to odorants [68]. Previous works show that benzyl salicylate can also have an enhanced role in the responses of ORs to specific odorants through allosteric modulation mechanism [89]. What’s more, the IMP amino group at Asn69 through hydrogen bond formation promotes the connection more tightly of the VFT active bag with ligands and facilitates intracellular signaling in responses of gustatory cells to monosodium L-glutamate according to previous research. The assistance of enhanced elements in a reaction system greatly enhances the detection performance of biomimetic biosensors responding to specific ligands through allosteric modulation mechanism. Therefore, the search and design of enhanced elements can be a potential trend in the improvement of detection performance of biomimetic biosensors construction for the identification and analysis of olfactory and gustatory molecular detection.

### 5.2. Sensitive Materials

Mammalian olfactory and gustatory systems can simultaneously identify and detect various odorant and tastant ligands with high sensitivity and specificity and wider response range, owing to the abundant diversity of functional elements of biological olfaction and gustation in mammals. Truly, infinitely imitating and approaching the functional characteristics of biological sensing of olfaction and gustation or even replacing their functional roles in the fields of medical, environmental safety, and food safety are the ultimate goals of many biomimetic biosensing platform construction. To greatly simulate the functional diversity of biological olfactory and gustatory systems, numerous biomimetic biosensors were widely developed based on more emphasis on the imitation effect of sensitive materials for complex odorant and tastant diversity detection, such as creative recombination of olfactory and gustatory functional elements including the co-culture of olfactory or gustatory functional cells with nerve cells [73], olfactory or gustatory tissue-engineered mixed culture of functional cells, recombination of multiple olfactory or gustatory functional proteins [85], and composition of protein-derived peptides designed and produced from protein functional domains.

Recently, the construction of organoids has provided the most efficient and potentially sensitive materials for the construction of biomimetic biosensing of olfaction [130] and gustation [131]. Organoids known as micro-organs with physiological functions, are three-dimensional assemblies derived from stem cells or tumor cells and their spatial patterns, structure, and function are approximate with the corresponding tissues or organs in vivo [132]. Organoids have preponderances of breaking through spatial limitations, better cell heterogeneity maintenance, self-renewal and self-organization abilities compared with two-dimensional cells, which have been widely used in tissue development and maintenance research [131], tumorigenesis and progression research [133], drug screening [134], toxicity testing research [135], and so on. In addition, the development of olfactory and gustatory organoid systems offers an effective platform in vitro for furtherly research on the mechanism of olfactory and gustatory coding. Notably, the culture system and physiological functions research of olfactory and gustatory organoids have emerged. Furthermore, the development of 3D bioprinting and 4D bioprinting structure based on compatible hydrogels as the main biomaterials to simulate the microenvironment in organisms provides potential development prospects for the stability and reproducibility of olfactory and gustatory organoids, as well as the function verification on transducer devices. For better monitoring and determining the variation of various physiological and biochemical parameters of olfactory and gustatory organoids and the changes in the surrounding environment caused by the complexity of odorant or tastant stimulation, the development of a co-culture system of olfactory or gustatory organoids with cells in the cellular microenvironment is necessary. In addition, an artificial cell structures construction based on biocompatible hydrogels was creatively proposed, greatly enriching the choice of sensitive materials, offering a potential direction for development of biomimetic olfactory and gustatory biosensors construction [136].

### 5.3. Detection Strategies

The smart portable devices construction has run through the development process of the entire sensor field, being the ultimate goal of different types of sensor development including a biomimetic biosensor for odorant and tastant detection. Its excellent characteristics including miniaturization, intelligence, excellent detection index (ultra-high specificity, sensitivity, repeatability, and reproducibility), allow for fast, accurate, and on-site detection, which have already been pursued by high-quality sensor research and development. Recently, the rapid development of microfluidic integrated devices has provided a potential platform for the intelligent and portable design and construction of biomimetic biosensors. The microfluidic technology with precise control and manipulation of micro-scale fluids characteristics integrates biological, chemical, and medical analysis processes on micro-scale chips and automatically completes the entire analysis process, which has been widely used in clinical diagnosis model construction in vitro [137,138], drug screening, and transport fields [139,140]. Its excellent characteristics of miniaturization, integration, automation, and high-throughput characteristics provide a potential research direction for achievement of complex and high-throughput detection of odorants or tastants. What’s more, the current rapidly developed digital microfluidics system with utmost integrated automated analysis and precise control of multiple parameters make it possible to dynamically manipulate and observe important physiological and pathological processes and activities of cells or tissues in real time. Simultaneously, the realization of organs and tissues cultured on microfluidics with the assistance of 3D biomimetic microstructures and biomaterials such as biocompatible hydrogels, make up for the limitations of traditional two-dimensional cell culture and animal experiments. The 3D cultural system renders certain the realization of physiological functions at the level of tissues and organs cultured in vitro, providing a reliability guarantee for the cultivation and functional verification of olfactory and gustatory organoids. 

Notably, the integrated combination of microfluidic devices with other functional detection platforms, such as electrochemical sensors [141] and an electrophysiological detection platform [142], has been extensively designed and developed for multifunctional evaluation and complex signal detection of various types of biomolecules. Recently, commercial portable electrochemical sensors have rapidly emerged and the designed disposable electrochemical electrodes can be embedded in many compatible platforms. Various research has successfully launched platforms based on the combination of the designed disposable electrochemical electrodes with microfluidic strategies to achieve high-sensitivity detection on small molecules. Prospectively, the successful combination of microfluidics system and multifunctional electrochemical sensors can greatly realize real-time monitoring and acquisition and signal feedback of fluctuant physiological and pathological multi-parameter detection at the tissue and organ level due to human intervention stimulus, which provides a powerful platform to explore the functional mechanisms of the olfaction and gustation system. In addition, the integrated automation biomimetic biosensors have been widely developed based on a microfluidics system as a culture microenvironment and a electrophysiological detection platform as a transducer part for real-time effective recording of the electrophysiological activity of excitable cells in many previous reports. 

However, the utilization of conventional two-dimensional planar electrodes may have a limitation in formation of the complete bio-interface system and measurement of high-quality and reliable electrophysiological recordings due to the existing multiple 3D structural features of different types of excitable tissues and organs. To challenge this dilemma, many studies have proposed and designed different types of 3D bioelectrodes system by changing the 3D geometry and microscopic size and lowering the Young’s modulus biocompatible materials to achieve a high resistance sealing and low impedance of the cell-electrode interval [143]. The integrated coupling of 3D bioelectrodes and a microfluidic device provides an efficient development platform for high-quality electrophysiology detection of excitable 3D structurally characterized olfactory and gustatory organoids and tissues when responding to different odorant and tastants stimuli, respectively. In addition, the integrated microfluidic system used as a biomimetic biosensor platform to verify the physiological functions of olfactory and gustatory system is required to realize the real-time recognition and classification of complex mixed odorants or tastants as well as real-time monitoring of and feedback on the physiological and pathological changes caused by odorant or tastant stimulation. Therefore, the high efficiency of a data processing system is very important for the realization of stable and sustainable operation of a microfluidic integrated system. The current prevalent artificial intelligence technology has been proposed to realize the efficient identification and classification of high-throughput data in microfluidic integrated systems. Koniku Corporation has developed a novel biomimetic olfactory biosensor based on the combination of a microfluidic equipped with several electrode array responding chambers and different types of olfactory-functional cells, such as genetically engineered nerve cells for detection of one or more different odorants. Combined with machine learning, it has successfully achieved automated odorant identification and offered an effective platform for hybrid odorant analysis. In terms of algorithm recognition, based on the architecture of the mammalian olfactory bulb, Intel Corporation utilized the algorithm of spike time to make use of spike timing-based algorithm utilizes distributed, event-driven computations and rapid (one shot) online learning to successfully realizes the rapid online learning and identification of odorants samples under interference [144]. Therefore, in the near future, it is possible for ultra-high performance biomimetic biosensors that mimic the mammalian olfactory and gustatory systems to be constructed, with the assistance of various algorithms, such as neural networks and various high-throughput array sensors, to quickly identify complex odorants and tastants signals, in conjunction with indicators, such as analogy pH indicator films [13] and genetically engineered phage-based color films [145,146], which can realize rapid detection and visualization of complex and mixed targets and can offer further development that is closer to the olfactory and gustation functional system of mammalian organisms. 

## 6. Conclusions

In conclusion, this review summarized the progress of biomimetic olfactory and gustatory sensing systems based on detection strategies in the past 10 years. In pursuit of olfactory and gustatory systems approaching that of the mammalian, abundant biomaterials were used as sensitive materials, including nanovesicles, nanodiscs, membrane receptor proteins, binding proteins, cells, and tissues for offering diverse options in the construction of biomimetic biosensors with high detection performance. In addition, the detection devices used as transducer elements, including FETs, MEAs, electrochemical sensors, QCM and optical devices, etc., have been widely used in olfaction and gustation detection and have provided a rich research background for portable development. 

## Figures and Tables

**Figure 1 biosensors-12-00858-f001:**
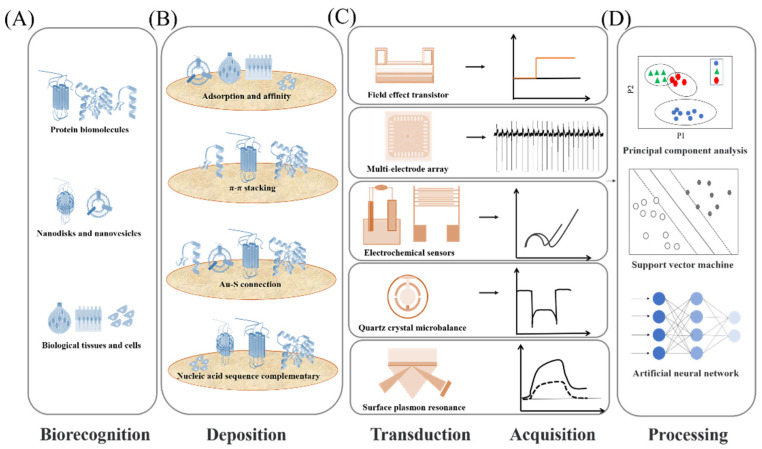
Scheme of main components construction of different detection strategies development for olfactory and gustatory recognition. (**A**) Biorecognition parts for the sensing of olfactory and gustatory ligands. (**B**) Effective deposition methods of sensitive materials on transducers surface. (**C**) Various transduction techniques to sense the responsive signals. (**D**) Signal processing and pattern recognition methods.

**Figure 2 biosensors-12-00858-f002:**
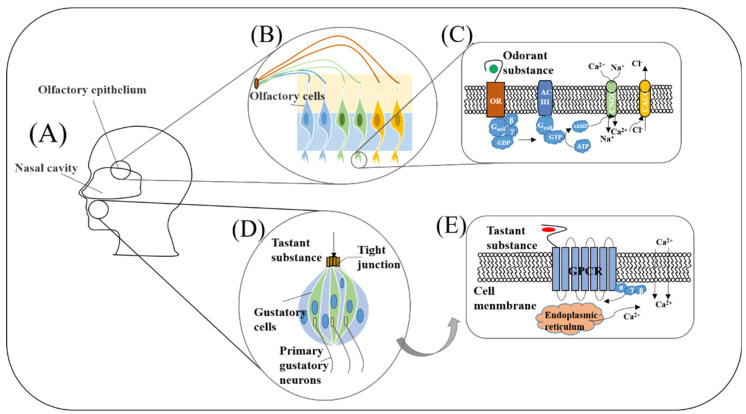
The overall structure of human olfactory (**A**–**C**) and gustatory systems (**A**,**D**,**E**).

**Figure 3 biosensors-12-00858-f003:**
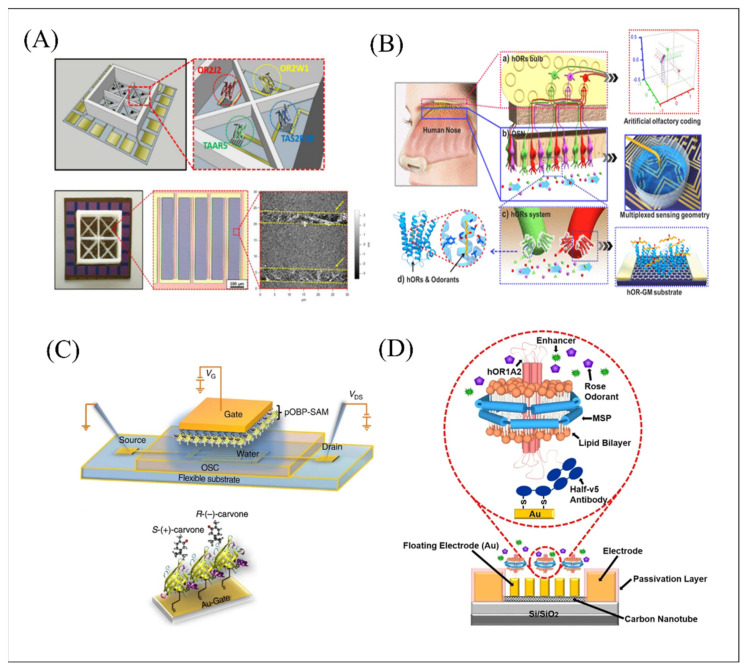
Schematic of biomimetic olfactory and gustatory biosensors based on FET functionalized with different types of sensitive materials. (**A**) Schematic diagram of multi-channel-type CNT-FET platform functionalized by human OR2J2, OR2W1, TAAR5, and TAS2R38. Reproduced with permission from Ref. [85]. (**B**) Schematic diagram of functional anatomy of human olfactory system and components of MSB-noses including graphene micropatterns and hORs for simulating each functional stage of the human nose. Reproduced with permission from Ref. [86]. (**C**) The schematic structure of WGOFET device functionalized with pOBP through a self-assembled monolayer (SAM). Reproduced with permission from Ref. [87]. (**D**) Schematic diagram of the floating electrode-based CNT-FET functionalized with nanodiscs embedded with hOR1A2 through half-v5 Ab fragments and thiol groups. Reproduced with permission from Ref. [89].

**Figure 4 biosensors-12-00858-f004:**
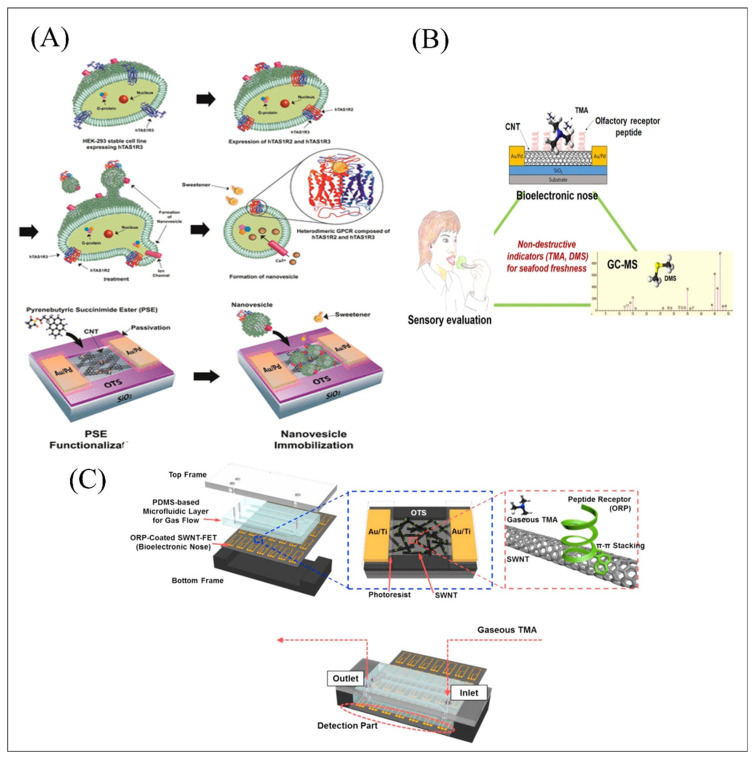
(**A**) Schematic diagram of the fabrication procedure of sweet taste biosensors functionalized with nanovesicles containing heterodimeric human sweet taste receptors composed of hTAS1R2 and hTAS1R3 through π-π interactions. Reproduced with permission from Ref. [75]. (**B**) A triangle study of a human evaluation test, instrumental analysis (GC-MS), and a CNT-FET bioelectronic nose functionalized with olfactory receptors for exploring non-destructive sensing of seafood freshness. Reproduced with permission from Ref. [55]. (**C**) Schematics of SWNT-FETs combined with μBN system for TMA molecule detection. Reproduced with permission from Ref. [70].

**Figure 5 biosensors-12-00858-f005:**
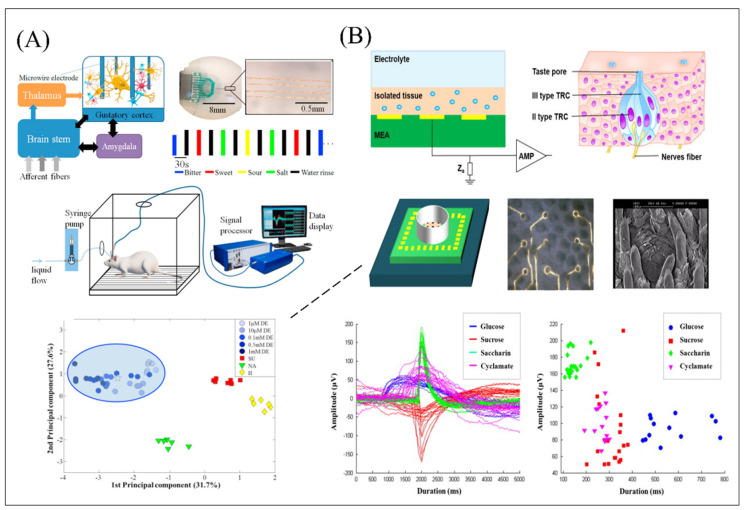
Schematic of biomimetic olfactory and gustatory biosensors based on electrode array functionalized with different types of sensitive materials and data analysis. (**A**) Description of gustatory signal pathway in rodents’ brain and the MEAs in vivo system coupling with specific neurons in GC for tastants detection, and the planar PCA plot with the first two principal components (PCs) accounting for 31.7% and 27.6% of the variance, respectively. Reproduced with permission from Ref. [102]. (**B**) Schematic of the microelectrode array functionalized with epithelium contained Type II taste cells for sweet detection and equipped with K-means algorithm and the signal-clustering pattern painting for data analysis. Reproduced with permission from Ref. [104].

**Figure 6 biosensors-12-00858-f006:**
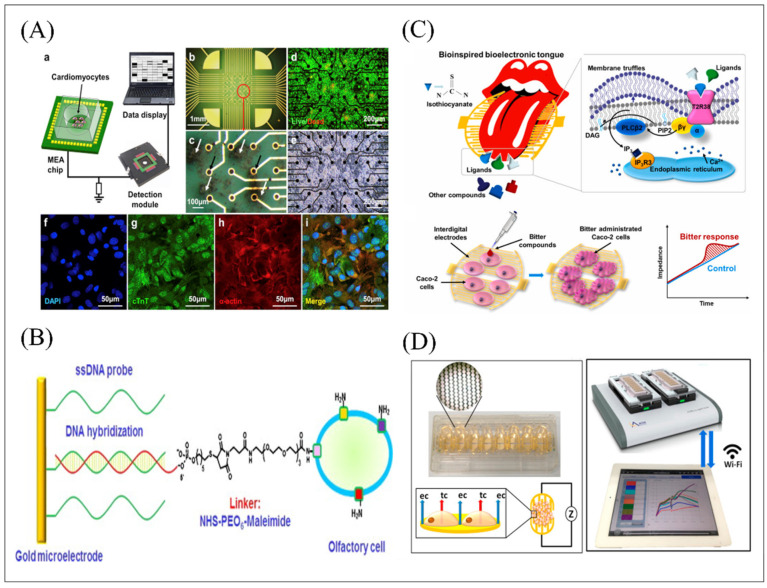
(**A**) The schematic diagram of MEA detection system coated with cardiomyocytes for tastants detection and related optical characterization (b–i). Reproduced with permission from Ref. [82]. (**B**) The schematic of MEA detection system functionalized with olfactory cells through Au-s connection formation and complementary ssDNA hybridization. Reproduced with permission from Ref. [106]. (**C**) Schematic diagram of the activation procedure of gustatory signal pathway and the detection principle of ECIS functionalized with Caco-2 cells expressing T2R38 for bitter compounds detection. Reproduced with permission from Ref. [107]. (**D**) Description of the ECIS recording system functionalized with male mouse germ cells expressing receptor T2Rs for bitter compounds detection and equipped with the iPad controls the recording and setting through Wi-Fi signal. Reproduced with permission from Ref. [108].

**Figure 7 biosensors-12-00858-f007:**
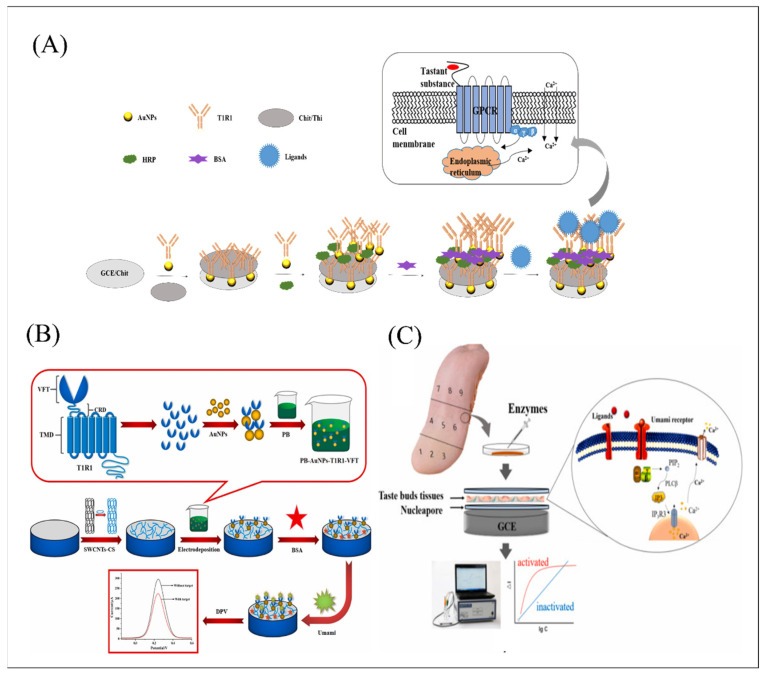
Schematic of biomimetic olfactory and gustatory biosensors based on electrochemical biosensor functionalized with different types of sensitive materials and data analysis. (**A**) Schematic of the double-layer gold nanoparticle (AuNPs) biosensor for simulating the linkage of the signal during the generation and amplification process from the extracellular to the intracellular sites on umami discrimination (right insert). Reproduced with permission from Ref. [66]. (**B**) Schematic of the preparation process of the T1R1-VFT electrochemical biosensor for umami detection. Reproduced with permission from Ref. [112]. (**C**). Schematic of the deigned gustatory biosensors based on sandwich-type structure consisting of aldehyde starch gel solution and sodium alginate solution in proportion and the different position of taste bud tissue as a core for umami detection. Reproduced with permission from Ref. [65].

**Figure 8 biosensors-12-00858-f008:**
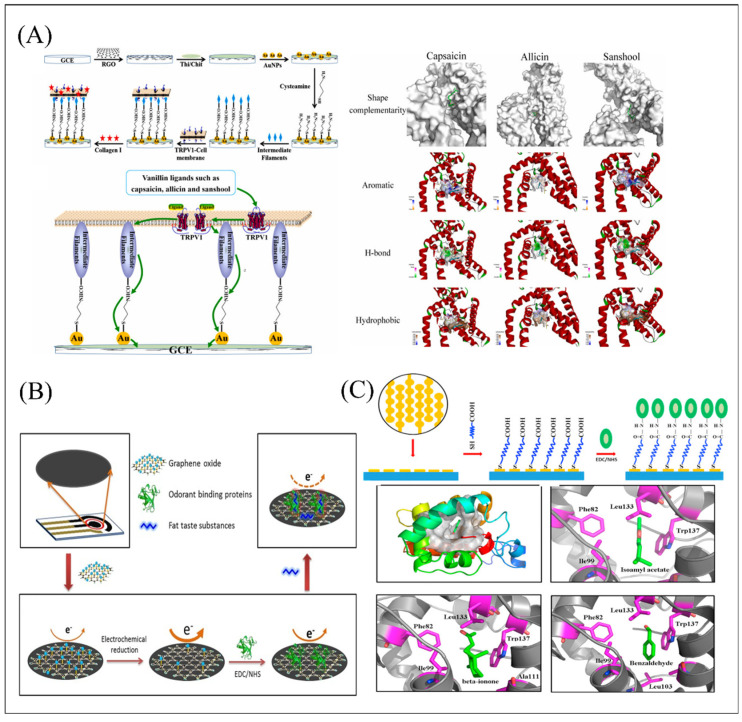
(**A**) Description of the preparation procedure, detection schematic diagram and molecular docking analysis of hTRPV1-cell membrane biosensor. Reproduced with permission from Ref. [67]. (**B**) Schematic diagram of the electrochemical gustatory biosensor based on screen-printed electrodes (SPE) for fat taste detection. Reproduced with permission from Ref. [115]. (**C**) Schematic of the preparation process of the interdigitated electrodes modified with PEG via Au-S bonds and functionalized with OBPs through EDC/NHS coupling and the molecular docking between BdorOBP2 and semiochemicals for affinity assessment. Reproduced with permission from Ref. [116].

**Figure 9 biosensors-12-00858-f009:**
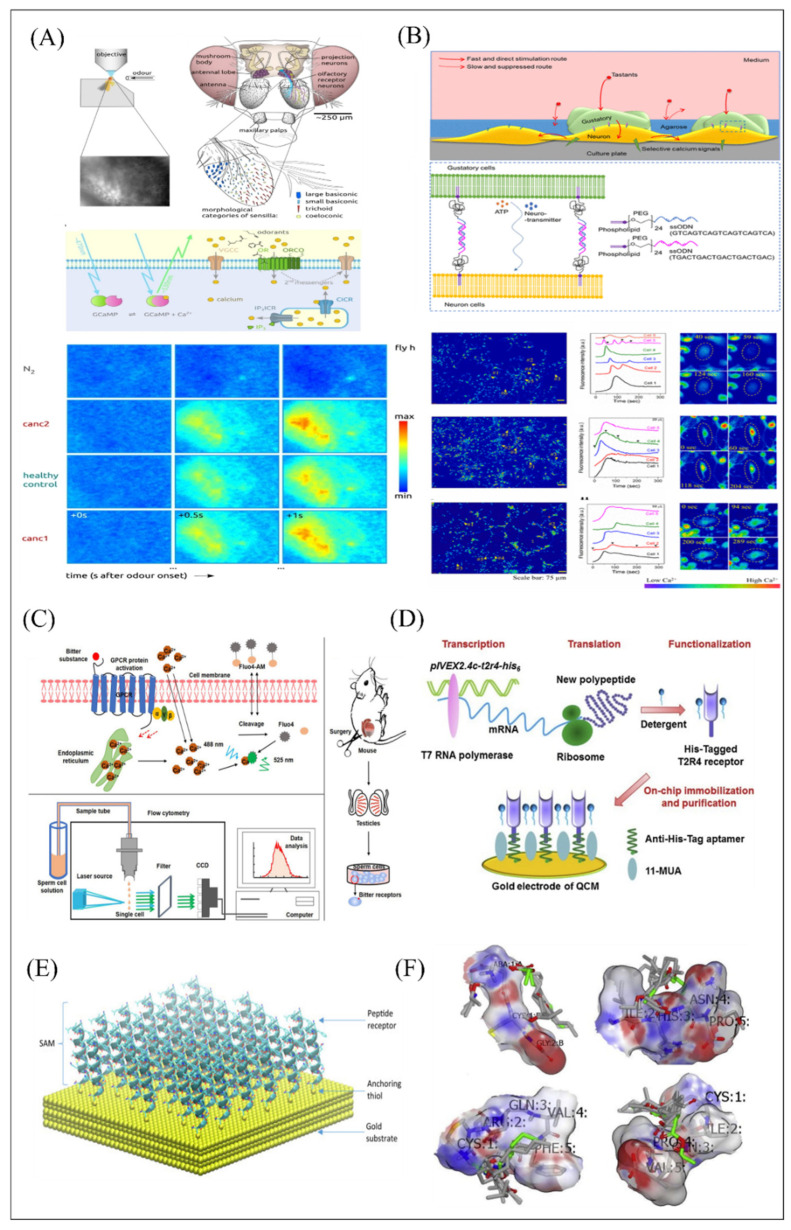
Schematic of biomimetic olfactory and gustatory biosensors based on fluorescence measurement. (**A**–**C**) and QCM (**D**–**F**) functionalized with different types of sensitive materials and data analysis. (**A**) Schematic mechanism of the calcium imaging principle of the fly’s antenna based on the combination of GCaMP with the increasing Ca^2+^ sources induced by odorant stimulus and the spatially different odorant response patterns showed that fluorescence changes in response to odorant stimulation differ for different cancer cell lines. Reproduced with permission from Ref. [120]. (**B**) Schematic illustration of the co-culture system of gustatory-neuron cells with gustatory cells through complementary oligonucleotides for selective taste simulation and the calcium imaging for function verification and concentrations optimizing the agarose thin layer of the co-culture system. Reproduced with permission from Ref. [74]. (**C**) The schematic diagram of sperm-cell-based biosensor (SCB) combined with the flow cytometry measurement system for fluorescent signals recording induced by bitter compound stimulus. Reproduced with permission from Ref. [122]. (**D**) Schematic process of His6-tagged T2R4 receptor production in a cell-free expression system and the QCM-based biomimetic taste biosensor construction. Reproduced with permission from Ref. [124]. (**E**) Schematic representation of biomimetic biosensor based on QCM functionalized with peptide molecules and high densely planting for long-chain aldehyde detection. Reproduced with permission from Ref. [72]. (**F**) The molecular docking results of glutathione, pentapeptides (CIHNP, CIQPV, CRQVF) with all volatile compound ligands. Reproduced with permission from Ref. [79].

**Table 1 biosensors-12-00858-t001:** Summary researches of detection strategies on biomimetic olfactory and gustatory biosensors based on field effect-based devices.

Transducer	Sensitive Materials	Detection Targets	Detection Limit/Detection Range	Selectivity	Ligand and Receptor Binding Index	Applications	Ref.
swCNT-FET	hOR2AG1	Amyl butyrate	1 fM/1 fM to 10 pM	Pentyl valerate, butyl butyrate, and propyl butyrate	-	Food industry	[83]
CNT-FET	The venus flytrap domain originating from T1R2	Sucrose and saccharin	0.1 fM/0.1 fM to 1 μM	Cyclamate, tasteless disaccharide, l-monosodium glutamate and denatonium	*K_d_*: 2.05 × 10^–11^ M and 6.88 × 10^–12^ M	Food safety	[84]
Graphene- FET	The venus flytrap domain originating from T1R1	l-monosodium glutamate	1 nM/1 nM to 10 µM	L-glutamine, sucrose denatonium benzoate and cyclamate	-	Food safety	[77]
Multi-channel swCNT-FET	OR2J2, OR2W1 TAAR5 and TAS2R38	Octanol, hexanol, trimethylamine, goitrin and trimethylamine, respectively	pM/100 fM to 100 nM		*K_d_*: 7.53 × 10^12^ M^−1^, 3.05 × 10^10^ M^−1^, 2.31 × 10^11^ M^−1^, 1.97 × 10^11^ M^−1^, 1.782 × 10^12^ M^−1^, respectively	Food safety	[85]
GMs-FET	hOR2AG1 and hOR3A1	Amyl butyrate and helional, respectively	0.1 fM/0.1 fM to 10 pM	hOR2AG1 (butyl butyrate, pentyl butyrate and hOR3A1 (Piperonal, safrole)	*K_d_*: 2.93 × 10^15^ M^−1^, 9.6 × 10^14^ M^−1^, respectively.	Spice in food industry	[86]
WGOFET	The monomeric porcine OBPs	(S)-(+)-carvone, (R)-(−)-carvone	1.1 ± 0.5 kJ mol^−1^/1 to 1 µM	2-phenylethanol	The dissociation constants K: 0.50 ± 0.01 μM, 1.22 ± 0.05 μM and 40 μM, respectively	Food flavor	[87]
rGO-FET	OBP14	Homovanillic acid, eugenol, and methyl vanillate.	Approximately 100 nM to 200 μM, 100 nM to 100 mM and 1 µM to 100 mM, respectively.	Methyl eugenol	*K_d_*: 4 × 10^−6^ M, 4 × 10^−5^ M and 2 × 10^−5^ M, respectively	Fragrance for cosmetics	[76]
GMs-FET	Nanodiscs embedded with TAAR13c and TAAR13d	Cadaverine and putrescine	1 fM	NH_3_, trimethylamine and putrescine (TAAR13c); NH_3_, trimethylamine and cadaverine (TAAR13d)	The binding energy: −3.5 kcal/mol, −3.6 kcal/mol, respectively.	Food application	[15]
CNT-FET	Nanodiscs embedded with TAAR13c	Cadaverine	10 pM/10 pM to 10 µM	Diaminodecane, trimethylamine, ethanolamine, glutamine	*K_d_*: 3.63 × 10^11^ M^−1^	The spoiled foods	[88]
	Nanodiscs embedded with hOR1A2	Geraniol and citronellol	1 fM and 10 fM, respectively./1 fM to 1 µM and 10 fM to 10 µM, respectively.	Trimethylamine and amyl butyrate	*K_d_*: 8.37 × 10^11^ M^−1^ and 2.60 × 10^6^ M^−1^	Cosmetics industry	[89]
	Nanovesicles containing AmGr10	l-monosodium glutamate	100 pM/100 pM to 10 μM	Sucrose and phenylthiocarbamide	*K_d_*: 1.77 × 10^8^ M^−1^	The liquid food	[90]
	Nanovesicles containing OR8H2	1-octen-3-ol	1 fM/1 fM to 1 nM	1-octanol, 1-octanal, 3-octanol, and 1-octene	-	Food application	[91]
	Nanovesicles containing hTAS1R2	Sucrose	-/500 µM to 5 mM	Cellobiose and D-glucuronic acid	*K_d_*: 2.0 × 10^−3^ M	Food application	[75]
	Nanovesicles containinghOR2AG1 fused with a Kir6.2 channel	Amyl butyrate	1 fM/1 fM to 100 fM	Butyl butyrate, pentyl valerate, propyl butyrate, hexanal and decanal	-	Spice in food industry	[92]
	The micelle-stabilized olfactory receptor from C. elegans	Diacetyl	10 fM/1 fM to 10 nM	2-butanone, 3-methyl-2-butanone	The equilibrium constant K: 1.7 × 10^−12^ M	Alcoholic beverages	[11]
	OBP-derived peptide	3-Methyl-1-butanol	1 fM/1 fM to 10 nM	2-methylbutane, methyl isopropyl ketone, 3-methyl-1-butanethiol, isobutyl acetate, 3-methylbutanal and 3-methylbutanoic acid.	*K_d_*: 5.25 × 10^13^ M^−1^	The Salmonella-contaminated food	[71]
CNT-FET	horp61m	Trimethylamine	10 fM/10 fM to 1 µM	Dimethyl sulfide	*K_d_*: 3.33 × 10^12^ M^−1^	The spoiled foods	[55]
CNT-FETs combined with μBN	The olfactory receptor-derived peptides	Gaseous Trimethylamine	10 ppt/1 ppt to 10 ppb	Triethylamine, dimethylamine, 2-methyl-1-propanol, acetic acid, and acetone solutions	-	The spoiled foods	[70]
Electrolyte-insulator-semiconductor	olfactory receptor ODR-10	Diacetyl	0.01 nM/0.01 nM to 1 nM	Butanone, 2, 3-pentanedione, and isopentyl acetate	-	Alcoholic fermentation	[93]
LAPS	taste bud cells from SD rats	Carbenoxolone	-	-	-	Drug screening	[94]
ITO-based electrolyte-semiconductor	hT2R4 expressed in *E. coli*	Denatonium	-/50 nM to 500 nM	Quinine and alpha-naphthylthiourea	-	Drug screening	[95]

**Table 4 biosensors-12-00858-t004:** Summary research of detection strategies on biomimetic olfactory and gustatory biosensors based on Ca^2+^ imaging and QCM.

Detection Methods	Sensitive Material Types	Target Compound(s)	Detection Limit or Minimum Detection/Detection Range	Selectivity	Potential Applications Fields	Ref.
The Ca^2+^ imaging in vivo based on GCaMP3	The fruit fly’s antenna	The volatile compound generated from cancer cells	-	1-butanol	Auxiliary diagnosis	[120]
The Ca^2+^ imaging based on Fluo 4-AM	The co-culture of taste and neuronal cells	Denatonium benzoate	1 mM; 5 mM	-	Drug screening	[73,74]
The Ca^2+^ imaging based on Fluo 4-AM and flow cytometry.	The functional taste cells	NaCl, sweet glycine, saccharin, and denatonium benzoate	100 mM, 200 mM, 50 mM 100 μm and 5 μM, respectively.	-	Drug screening and food application	[80]
The Ca^2+^ imaging based on Fluo 4-AM and microfluidic platform	Human embryonic kidney-293 cells expressing hOR1D2, hOR1A1 and hOR1G1	Bourgeonal, β-citronellol, and geraniol	0 to 2 ppm	β-citronellol (hOR1D2), bourgeonal and geraniol (hOR1A1), geraniol (hOR1G1)	Spices in cosmetics	[81]
The Ca^2+^ imaging based on GCaMP3	Sf21 cells expressing odorant receptors, Orco, and GCaMP3	Bombykol and bombykal	1 µM (238 ppb) and 300 nM (70.9 ppb), respectively/∼300 nM to 30 µM	Bobbykol (BmOR1), bobbykal (BmOR3)	Environmental conservation	[121]
The Ca^2+^ imaging based on Fluo 4-AM and flow cytometry.	The Mouse germ cell	Denatonium benzoate, N-phenylthiourea and quinine.	15.6 μM, 25.1 μM and 25.1 μM, respectively/50 to 4000 μM, 25 to 1000 μM and 50 to 2000 μM, respectively.	Citric acid, sucrose, monosodium glutamate and NaCl	Drug screening	[122]
QCM based on the gold electrode for frequency measurement	*C. elegances*, ODR-10	Diacetyl	1.5 ppm (*v*/*v*)/10 ppm to 100 ppm	Isoamyl acetate, anisole, lavender, butanone, and 2,3-pentanedione	Alcoholic beverages	[123]
	T2R4	Denatonium	5 nM/10 nM to 0.1 mM	MgSO4, D- (-)-salicin, and quinine	Drug screening	[124]
	The peptide mimicking HarmOBP7 region (KLLFDSLTDLKKKMSEC)	Nonanal and so on (more than 5 carbon atoms in the structure)	14 ppm/42 ppm to 1303 ppm	Formaldehyde, acetaldehyde, glyoxal, benzaldehyde, propanal, hexanal and so on	Flavoring agent	[72]
	Five different peptides (cysteinylglycine, glutathione, CIHNP, CIQPV and CRQVF)	14 volatile compounds	-	-	Environmental conservation	[79]
	Five different peptides (IHRIC, KSDSC, LGFDC, TGKFC and WHVSC)	13 volatile compounds tested based on their hydrophobicity and hydrophilicity	-	-	Environmental conservation	[78]
SPR for a variation of refractive index detection	Liposomes containing hOR3A1	Helional	10^−7^ M/10^−3^ M to 10^−7^ M	Safrole, piperonal, hydrocinnam aldehyde, 3-(3,4-methylene diozyphenyl) propionic acid	Spices in cosmetics	[125]
SPRi	Three derivatives of OBP3	β-ionone and hexanal	200 pM and 100 g/mol, respectively./100 pM to 1 nM and -.	Hexanoic acid	Flavoring agent	[63]
SWCNTs-SPR	TNT-binding peptide (ARGYSSFIYWFFDFC)	2,4,6-trinitrotoluene	772 ppb/0.8 to 12.5 ppm	DNP-glycine, 2,6-DNT, research development eXplosive, and 4-nitrobenzoyl-glycyl-glycine	Security application	[18]
SPR	Three TNT binding peptide sequences (ARGYSSFIYWFFDF)	2,4,6-trinitrotoluene	3.4 ppm/4.0 ppm-250.8 ppm	TNP-glycine, DNP-glycine, 2,4-DNT, 2.6-DNT, 4-Nitrobenzoyl-glycyl-glycine, research development explosive	Security application	[56]

**Table 5 biosensors-12-00858-t005:** Comparison of different biomimetic strategies based on existing research strategies.

Detection Strategies		Sensitive Materials Types	Advantages	Disadvantages	Portable Development Possibilities	Most Applications
Surface functional FET for conductivity changes		Receptors, binding protein, artificial peptide, functional nanovesicles and nanodiscs	Ultra-high sensitivity	Immobilization distance	High	Food security
MEAs for electrophysiological detection	In vivo	Related functional parts of the brain	Most genuine response	Invasive, requirement of trained specialists	Moderate	Detection of explosives
	In vitro	Functional cell, tissues, organoids	Non-invasive	Time-consuming process	Low	High throughput screening
Electrochemical method		Receptors, binding protein, artificial peptide	Portable, Robustness	Sensitive to sample matrix	High	Affinity assessment
QCM for frequency change detection		Receptors, binding protein, artificial peptide	Stable output	Interference induced by nonspecific binding	High	Screening peptide function verification
Optical measurement	Ca^2+^ imaging	Functional cell, tissues, organoids	Real-time detection	Low reproducibility	Low	Live recording
	SPR imaging	Binding protein, artificial peptide, functional liposomes	Simultaneously monitoring large sensor arrays in real time	Interference induced by nonspecific binding	High	High throughput peptide screening

## Data Availability

Not applicable.

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
