# Peer review of "Progress in the Development of Detection Strategies Based on Olfactory and Gustatory Biomimetic Biosensors"

_biosensors, 2022, doi:10.3390/bios12100858_

Round 1

Reviewer 1 Report

This review is an overview of the manuscript submitted by Chen et. al.  but the main critical points are underlined.

The subject of the review article written by chen et al. is of interest.  It covers a wide range of the literature dealing with olfactory and gustatory biomimetic sensors.

Following an introduction around the sensing mechanisms and  immobilization strategies, the article focuses on the transducers (part 4). This organization of the part 4 generates redundancy concerning the description of the sensing parts ie proteins, peptides, receptors, cells… making the manuscript difficult to read.

It would be more efficient and probably more interesting to organize the ms around the different biomimetic approaches.

The authors want to include gustatory and olfaction processes in this ms; there is no specific mention of the type of samples : liquid phase or gas phase.  These data are merge sometimes in one table ex on table 4 where we can find liquid solutions (concentration in M), or gaseous-volatil samples (concentration in ppm, or ppmv? )  . The specifications of the sensor construction are completely different for these two applications regarding the problem of stability of the biological materials on a dry sample. It could be a good idea to split the sections into 2 parts devoted to gas (olfaction)  and liquid samples for taste analysis…

The ms does not mention optical transducing processes using biomimetic approaches (SPRimaging, Mach Zehnder interferometers)…or biomimetic devices constructed and commercialized by private companies ie Koniku, Aryballe… scientific publication can be found on these topics.

The LOD seems to be the major parameter analyzed, there’s no specific comments about the specificity of the described sensors. This remains a major point since the aqueous or gaseous samples are always complex mixtures. Can the authers comment this point regarding the cited references.

All these points are suggestions to improve the readability and the interest of the manuscript but in a overall point of view it merges a very important quantity of data. The main problem remains the organization of these data connected to the probably too large scope of the review.  

Minor points:

Line 50 : the reference 2 deals with “Security applications”, and bioterrorism not “Social security” or  and “maintening social stability”.  The manuscript must not be linked with specific Chinese policy.

Figure 1 : change “biorecocognition” by receptors

Line 145 : physiological description of the mechanisms of olfaction and gustation is very ambitious,  perhaps separate the 2 parts in section 2.2 and introduce in 5 line the major role of OBP for olfaction.

Table 1 please avoid abbreviations of the target compounds AB, MSG…

Line 163 “fresh” is not a basic test. Saverness or umami is missing (see Wikipedia)

Fig 4 i/5 are very small!

Reviewer 2 Report

Comments in the attached file

Round 2

Reviewer 2 Report

Accept